# Relational visual representations underlie human social interaction recognition

Manasi Malik ®[1] ✉ & Leyla Isik ®[1] ✉

Humans effortlessly recognize social interactions from visual input. Attempts to model this ability have typically relied on generative inverse planning models, which make predictions by inverting a generative model of agents' interactions based on their inferred goals, suggesting humans use a similar process of mental inference to recognize interactions. However, growing behavioral and neuroscience evidence suggests that recognizing social interactions is a visual process, separate from complex mental state inference. Yet despite their success in other domains, visual neural network models have been unable to reproduce human-like interaction recognition. We hypothesize that humans rely on relational visual information in particular, and develop a relational, graph neural network model, SocialGNN. Unlike prior models, SocialGNN accurately predicts human interaction judgments across both animated and natural videos. These results suggest that humans can make complex social interaction judgments without an explicit model of the social and physical world, and that structured, relational visual representations are key to this behavior.

Humans easily make social evaluations from visual input, such as deciding whether two agents are interacting or who is a friend versus a foe. Early work by Heider and Simmel (1944) showed that humans can recognize rich information about others' interactions even from simple visual cues. The ability to distinguish between helping versus hindering individuals in visual displays has even been found in infants[1,2] and non-human primates[3]. Attempts to model human social interaction judgments typically rely on generative inverse planning models[4–7]. These models recognize social relationships by inverting a generative model of agent's interactions and comparing an observed social scene to internally generated hypotheses based on not only visual information, but also extensive physical information about the scene and hypothesized goals of the agents. So far, these models provide the best match to human judgments, suggesting humans rely on similar inferential processes to recognize interactions. In their current instantiations, however, these models are computationally expensive and often intractable in real-world scenes where a full physical simulation is infeasible (though it is possible to implement these systems in a tractable manner if the correct situational constraints are known[8]).

More generally, inverse planning models necessitate the use of explicit mental representations of other agents' minds and the physical world to make social judgements. While humans can clearly use high-level mental state inference to recognize and understand many aspects of social interactions[6,9,10], especially when visual cues are non-diagnostic, growing evidence suggests that social interactions are also rapidly recognized by the human visual system[11]. Social interactions have been shown to receive priority access to conscious awareness in a binocular rivalry task[12], provide a perceptual advantage in visual search tasks[13,14] and, like faces, be subject to an inversion effect[15]. Beyond detecting social interactions, humans also rapidly and automatically encode event roles, such as who acted on whom, in scenes depicting social interactions[16]. Neuroimaging evidence suggests that interacting dyads are represented in visual cortex[13], and are selectively processed in brain areas separate from those associated with theory-of-mind or mental state inference[15,17,18,14]. Despite this evidence, the field still lacks good bottom-up, visual models of social interaction recognition. Even deep learning models that achieve human-level performance in so many other visual tasks do a poor job modeling human social interaction judgments[4,19]. It thus remains an open question to what extent

[1]Department of Cognitive Science, Johns Hopkins University, Baltimore, MD 21218, USA. ✉e-mail: mmalik16@jhu.edu; lisik@jhu.edu

humans rely on visual processes versus higher-level cognitive reasoning for social interaction recognition. (Note here that by visual we do not mean that humans exploit very simple pixel-level information to recognize interactions. Nor, on the other hand, do we mean visual information that is processed by higher-level cognitive systems. Instead, the above evidence suggests that like other high-level features, including causality and animacy[20], social interactions are also processed within the visual system using spatiotemporal cues in a manner that is distinct from cognitive processing[21].)

A key property of social interactions is the fact that they are relational. In order to recognize an interaction between people, you not only need to recognize the body, pose, and motion of each person, but the relative distance, position, and motion between them[22–24]. We exploit this property and introduce it as an inductive bias, in the form of a graph structure[25], to a bottom-up visual neural network model. The result is a graph neural network (GNN) for social interaction prediction that we call SocialGNN, which serves as a representational level model of human social perception. We find that SocialGNN matches social interaction judgments at the level of human agreement on a dataset of animated shape videos, performs significantly better than a matched neural network model without graph structure, and is on par with generative inverse planning models. Unlike prior models, SocialGNN can predict human judgments in both animated and natural videos without explicit representations of the agents' mental states. These findings provide important insight into how visual information is used for human social interaction judgments.

## Results
### Humans recognize interactions and attribute sociality to those interactions in animated shape videos
To investigate human social interaction judgments, we first used the PHASE dataset[26]. This dataset consists of 500 animated shape videos in the style of Heider and Simmel, where two agents are moving around in a simple environment, resembling real-life social interactions (see example screenshot in Fig. 1). The dataset includes a 400-video standard dataset, and a 100-video generalization set with novel environment layouts and agents with social and physical goals that are unseen

in the original 400-video set. Each video has a ground truth label for the social interaction type between the two agents—"friendly", "neutral", or, "adversarial"—based on the agents' pre-defined social goals provided to the simulator during video generation. To compare these labels to human behavior, we collected human judgments of the social interaction depicted in each video (Fig. 1, Supplementary Fig. 1). After cleaning responses, we had at least 10 ratings per video, from 308 participants.

For each video, we compared the mode of the human ratings to the social interaction label (i.e., the label derived from the social goal used to render each video) from the PHASE dataset (Table 1). We find that videos generated with agents having "friendly" or "adversarial" goals were typically rated as similarly "friendly" or "adversarial" interactions by our participants. However, videos rendered to depict "neutral" goals were often rated as "friendly" or "adversarial" instead. This suggests that humans tended to recognize interactions between agents in a scene and attributed sociality to those interactions. For example, if both agents had a neutral goal of taking the blue ball to the green landmark, that may be perceived as a friendly, collaborative interaction. This highlights an important distinction between human judgments and the simulator used to produce the PHASE videos, as well as the importance of using human ratings rather than default labels from the simulator.

### SocialGNN predicts human social interaction judgments in animated shape videos
We developed a graph neural network, SocialGNN, to predict social interactions in the PHASE videos. For each video, SocialGNN takes in a graph representation of the visuospatial information for each frame. The nodes in these input graphs are the entities (agents and objects) in the frame and the edges represent relationships between the entities, thereby incorporating a relational inductive bias (Fig. 2a). For each video, this sequence of graphs now acts as the input to SocialGNN.

The overall network architecture (Fig. 3a) is similar to a recurrent neural network (RNN) that at each time step processes new input information ($\mathbf{G_{in}}^t$) and combines it with the learned representations from prior timesteps (Gtemporal module). It combines these

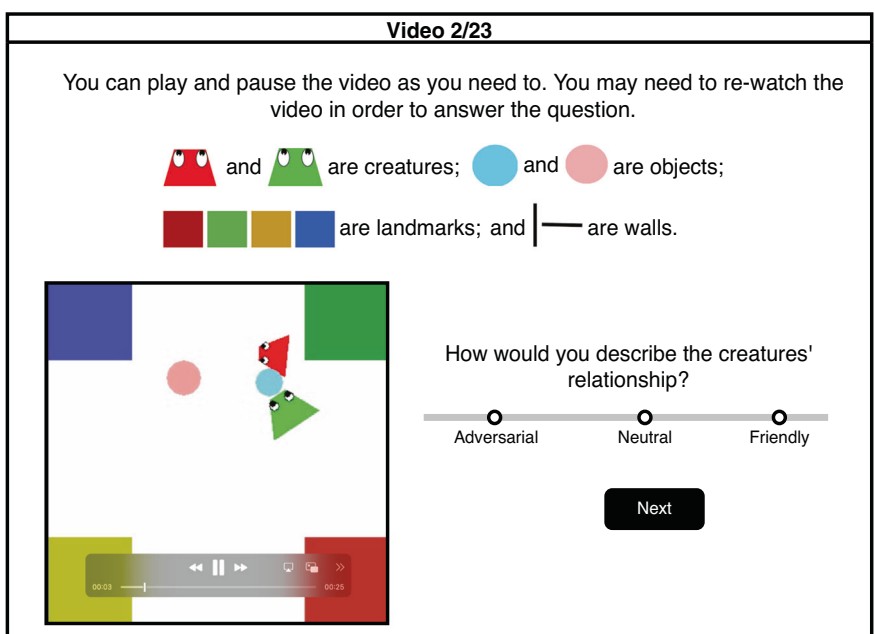

**Fig. 1 | Example display from the online human subjects experiment conducted to collect social interaction ratings for PHASE animated shape videos.** Full instructions were presented at the beginning of the experiment (Supplementary Fig. 1) with abbreviated instructions above each video. The participants viewed the videos and rated each video as "friendly", "neutral", or "adversarial".

representations across time and uses the representation at the final time step to predict the type of social interaction as "friendly", "neutral", or "adversarial" via a linear classifier. The novelty of this model comes from its graph structure (Fig. 3b). The input node features ($\mathbf{V_{inp}}^t$) are various visual and spatial properties of the entities, including 2D position, 2D velocity, angle, size, and whether that entity is an agent or object. Nodes in the input graph are connected via edges ($\mathbf{E_{inp}}^t$). A binary, bidirectional edge is added between the two entities if they are in physical contact (Fig. 2a). The input graphs also have some context information in the form of wall and landmark positions. Importantly, all input features and edges can be extracted from purely visual input.

We compared SocialGNN to a baseline model with the same overall architecture and input, but without the graph structure, called VisualRNN (Fig. 3c). Here, instead of the graph input and processing, an RNN with long short-term memory (LSTM[27]) gets concatenated features of all the entities in the scene at each time step and combines this input with learned representations from prior time steps. At each time step, wall and landmark coordinates are also concatenated with entity features to provide the model with contextual information. VisualRNN is an implementation of a Cue-Based LSTM, a standard perceptual baseline used in similar tasks[26,28]. We also compared

SocialGNN with an enhanced control model: VisualRNN-Rel. For VisualRNN-Rel, along with the entity and context features, we append a relational input feature (boolean vector denoting which edges are present) to match the binary edge information provided to the GNN. Finally, we also compared SocialGNN and the baselines to the performance of an instantiation of an Inverse Planning model, SIMPLE, which currently achieves state-of-the-art performance on this task[26]. The Inverse Planning model has access to all input information provided to SocialGNN and VisualRNN models via its built-in physics simulator and input state information.

We trained SocialGNN and baseline models on ten bootstrapped train/test splits of 300/100 videos from the standard PHASE dataset, using the human judgments as ground truth. For the Inverse Planning model, we got the predictions for all videos and calculated the performance accuracy for each bootstrap. We found that SocialGNN predicted social interaction judgements significantly above chance, almost reaching the level of human agreement among the raters (Fig. 4, Supplementary Fig. 7). SocialGNN performed significantly better than the matched visual model (VisualRNN), suggesting that relational graphical representations allow bottom-up visual models to make more human-like social judgments (paired permutation test based on exact null distribution $n = 1024$, $p = 0.002$). Interestingly, simply adding relational information as an input feature to the VisualRNN (VisualRNN-Rel) did not improve its performance. SocialGNN even outperformed the Inverse planning model on the standard PHASE dataset ($p = 0.002$). Via an ablation study (Supplementary Fig. 4), we found that if the agent-object edges are ablated, SocialGNN's performance suffers even when agent-agent edges are left intact, suggesting that the agent-object interaction information is often essential for understanding the social interaction between the two agents. Together, these results suggest that relational information and graphical representations, in particular, are important and sufficient for human-like social interaction recognition.

**Table 1 | Human social interaction ratings vs. PHASE ground truth social goals (500 videos)**

|  |  | PHASE Labels | | |
|---|---|---|---|---|
|  |  | Friendly | Neutral | Adversarial |
| Human Ratings | Friendly | 114 | 80 | 4 |
|  | Neutral | 12 | 96 | 4 |
|  | Adversarial | 3 | 120 | 67 |

Confusion matrix showing agreement between the mode of human ratings (y-axis) and PHASE generator labels (x-axis) for 308 participants.

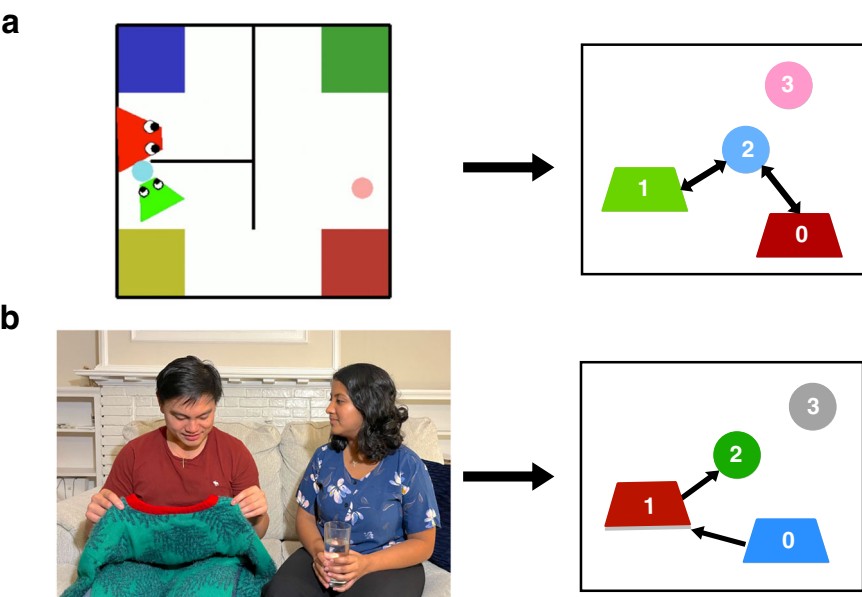

**Fig. 2 | Examples of how a graph (right) is created for a single frame (left) in a video for the PHASE and Human Gaze Communication datasets. a** For the PHASE dataset edges are determined based on physical contact. For this example, we see that the frame is represented as a graph using 4 nodes corresponding to the 2 agents (red and green) and the 2 objects (pink and blue). Since the green agent and red agents are touching the blue object, we add bidirectional edges between green and blue nodes, and red and blue nodes. **b** For the Gaze dataset, edges are based on gaze direction. The woman in blue and the man in red, are represented via

blue and red nodes respectively. The objects in their hands (green sweater and the glass of water) are represented via the green and gray nodes. Since the woman is looking at the man, and the man is looking at the object in his hands, two directed edges are added from the blue to the red node and from the red to the green node. Because of license restrictions with the Gaze dataset videos, we use a representative image in (**b**). Written consent was obtained from individuals shown in the image to use the image in the paper.

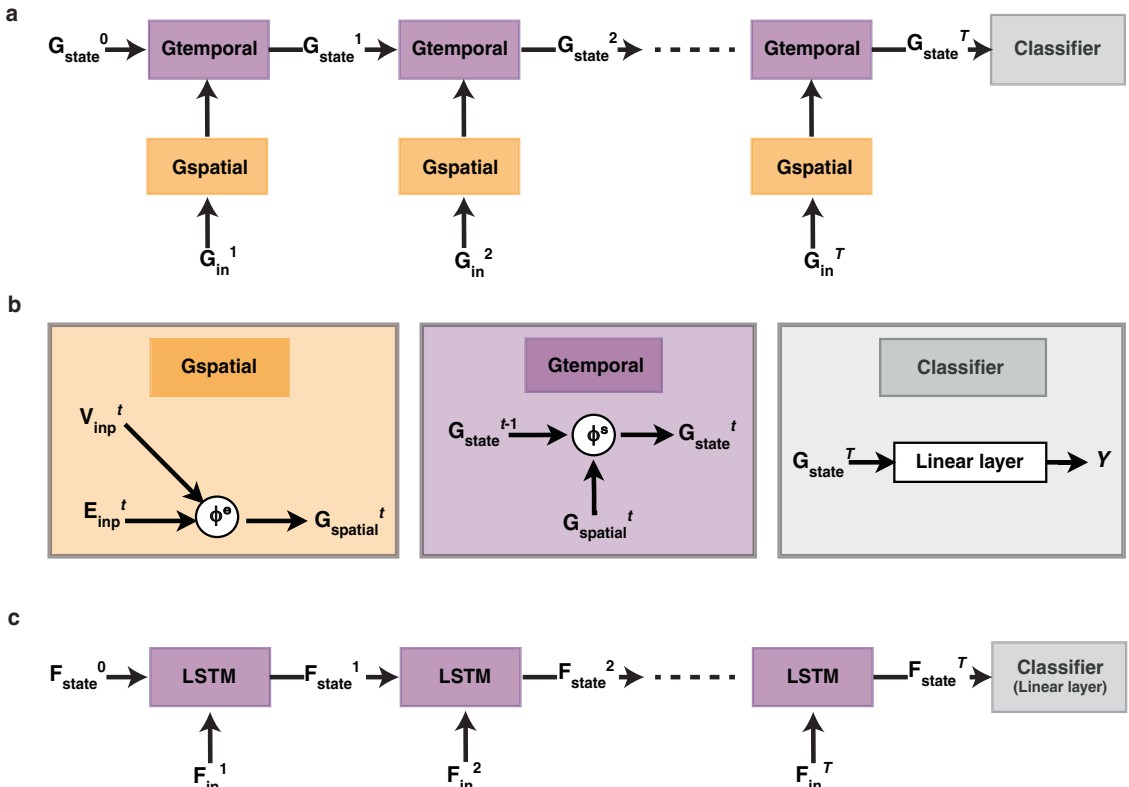

**Fig. 3 | Model architectures. a** Overall structure for SocialGNN. At each time step, the graph representation of the video frame ($G_{in}^t$) is passed to the Gspatial module. The Gspatial module processes the graph and outputs a vector representation, which is passed on to the Gtemporal module. Gtemporal module then combines information from previous time steps and the current time step. The representation at the final time step is passed through a linear classifier to predict the social interaction. **b** Details of the Gspatial, Gtemporal and Classifier modules of SocialGNN. $\phi^e$ in the Gspatial module is a linear layer, akin to those in a standard neural network, that receives as input the concatenated node features from nodes on each side of an edge (Supplementary Fig. 3a), and using these, along with context information, learns an updated representation for that edge. Updated representations for all the edges are concatenated ($G_{spatial}^t$) and then passed to the Gtemporal module where a long short-term memory (LSTM) unit (function $\phi^s$) combines information from previous time steps and the current time step to give a new state representation ($G_{state}^t$). The representation at the final time step ($G_{state}^T$) is passed through a linear classifier to predict the social interaction ($Y$). **c** Baseline model, VisualRNN, architecture. The structure is an RNN where features at each time step ($F_{in}^t$) are combined with the learned representations from previous time steps. At the final time step, a linear classifier is used to make a prediction about the social interactions.

## SocialGNN generalizes to novel visual and social scenes

To understand how well SocialGNN generalizes to novel environments and social actions, we trained SocialGNN and the visual control models using the standard 400 PHASE videos and tested them on the 100-video generalization videos. While the training and testing within the standard dataset was always done across novel videos, the physical environments and social actions were similar across those videos.

SocialGNN again performed better than the matched visual model with an accuracy approaching average human agreement (Fig. 5a, Supplementary Fig. 8). Unlike the standard PHASE dataset, here the Inverse Planning model performed better than SocialGNN. However, this performance comes at a large computational cost. The Inverse planning model requires thousands of times as much computing power (as measured by both run-time and memory, Fig. 5b) than either SocialGNN or VisualRNN.

## SocialGNN uniquely explains variance in human social interaction judgements

On the PHASE dataset, we found that both SocialGNN and the Inverse Planning model match human judgments of observed social interactions (Figs. 4, 5a). To investigate if the two models contain distinct or shared representations of the stimuli, we performed an item-wise comparison using Representational Similarity Analysis (RSA). We created Representational Dissimilarity Matrices (RDMs) for SocialGNN, the Inverse Planning model, and the human judgements. To construct

the RDMs, we used the output of the last RNN step for the SocialGNN representation, the predicted probabilities for each hypothesis for the Inverse Planning model representation, and the distribution of ratings per video for the human judgements' representation. We found that both SocialGNN and the Inverse Planning model were significantly correlated with human judgements (Standard Set: $\mu(r_{SocialGNN}) = 0.47$, $\mu(r_{InversePlanning}) = 0.25$, Generalization Set: $r_{SocialGNN} = 0.51$, $r_{InversePlanning} = 0.51$, $n = 10{,}000$, $p < 0.001$ for all) (Fig. 6, left). These results are in line with both models' high prediction performance and further suggest that the models match not only the majority label assigned by raters but also capture ambiguity in the human ratings (see Supplementary Videos 1–4 for examples of model agreement and disagreement with each other and human labels).

We also found a significant correlation between SocialGNN and the Inverse Planning model (Standard Set: $\mu(r) = .24$, Generalization Set: $r = .4$, $p < 0.001$ for both). To understand the extent to which each model captures unique variance in human behavior, we calculated semi-partial correlations with the human ratings RDM for each model while controlling for the other model. We found that both models capture significant unique variance in human judgements across both stimulus sets (Standard Set: $\mu(sr_{SocialGNN}) = .43$, $\mu(sr_{InversePlanning}) = .14$, Generalization Set: $sr_{SocialGNN} = 0.34$, $sr_{InversePlanning} = 0.33$, $p < 0.001$ for all) (Fig. 6, right). SocialGNN had a significantly higher correlation and semi-partial correlation on the Standard Set (Fig. 6a) ($p = 0.002$ for both), and both models performed similarly on the Generalizaton Set

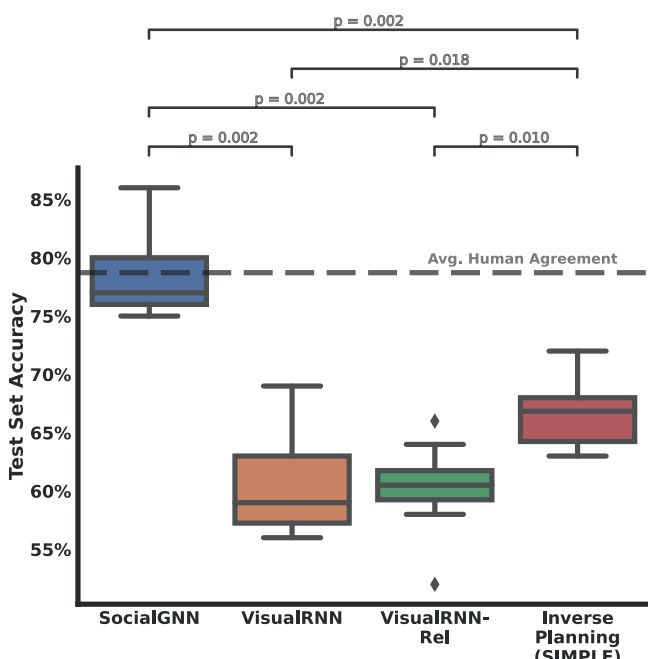

**Fig. 4 | Prediction accuracy for the type of social interaction (friendly vs. neutral vs. adversarial) between two agents in the PHASE dataset (chance 33%).** Box whisker plots showing each model's accuracy across the train-test splits. The center lines depict median accuracy, boxes middle 50%, diamond points the outliers, and whiskers accuracies outside the middle 50%. Significant differences are denoted by $p$-values (paired permutation test, $n = 1024$ permutations, all two-tailed, Holm-Bonferroni corrected).

(Fig. 6b). These results indicate that there is a significant amount of variance in human judgments that is uniquely explained by each SocialGNN and SIMPLE, and that the two models are learning distinct representations.

### SocialGNN recognizes social interactions in natural videos
In the real world, we watch people interact with each other and objects in dynamic, natural scenes. To model these real-world scenarios, we turn to the Human Gaze Communication dataset[29] (referred to as Gaze dataset henceforth). This dataset is a compilation of 299 videos from YouTube covering diverse social scenes. Each video is annotated with human face and object bounding boxes, and the heading/gaze direction of each face. The dataset included human annotations of different types of gaze communication, including "non-communicative", "mutual gaze", "gaze aversion", "gaze following", and "joint attention". We bootstrapped 20 train/test splits of 224/75 videos, and divided each video into shorter clips such that clips from the same video were either included in the train or the test set. After preprocessing, we were left with ~740 video clips for training and ~215 clips for testing in each bootstrap.

We again created graphs for each time step where nodes were made up of agents and objects, but here edges were directed and determined based on gaze direction (Fig. 2b). Additionally, to capture the rich, visual information in the scene, node features were generated by creating a bounding box around each entity and passing the pixel values through a deep neural network (VGG19 pre-trained on ImageNet[30]). Despite the drastically different input, the resulting graphs were of a similar structure to those used in the animated shape experiments and served as input to SocialGNN.

First, we trained the models to detect social interactions (presence vs. absence i.e., "non-communicative" vs. all other communicative gaze labels in the dataset) between two people in each video. The SocialGNN model performed significantly better than the

VisualRNN models (Fig. 7a, Supplementary Fig. 5) (paired permutation test, $p < 0.001$, $n = 10,000$ resamples). We next evaluated the model's match to behavior on a 5-way social interaction discrimination ("non-communicative", "mutual gaze", "gaze aversion", "gaze following", and "joint attention"). We again find that SocialGNN performs significantly better than the VisualRNN models (Fig. 7b, Supplementary Fig. 6) ($p < 0.001$). (Since the current Inverse Planning model is not an image-computable model, it cannot be included in these experiments.)

We also compared our model's performance with that of a standard visual CNN model, VGG19[30]. We extracted the penultimate layer representation from the frames of each video and trained a linear classifier on the above two-way and five-way tasks. The VGG19 model was significantly worse at matching human behavior in both tasks than SocialGNN ($p < 0.001$), performing at chance in the 2-way classification and slightly above chance in the 5-way classification, further emphasizing the benefits of relational structure for social interaction recognition.

## Discussion
We developed a computational model, SocialGNN, that reproduces human judgments of social interactions in both animated and natural videos using only visuospatial information and bottom-up computation. This model performs as well as a generative inverse planning model and does so at a fraction of the computational cost without any explicit mental inference of agents' goals, suggesting that computations within the visual system may be sufficient for humans to recognize social interactions. Interestingly, SocialGNN is not simply a neural instantiation of the inverse planning model, and both models explain unique variance in human judgements, suggesting humans are using a combination of perceptual and mentalistic strategies to judge these videos.

The SocialGNN architecture can also generalize across vastly different stimulus sets: one set of animated videos where social interactions are based on motion trajectories and physical contact, and a second naturalistic dataset of human gaze communication with minimal motion. The graph representations and computations allow SocialGNN to abstract away from low-level visual cues, but to do so in a way that is still visually grounded and can thus operate on natural videos. We note though that while SocialGNN performs significantly above chance on both natural video experiments, it is still far from the level of human performance in this domain. It seems likely that better engineered node features (image-level information about the agents and objects in each video) would improve the performance of all visual models. Optimizing these features is clearly an important engineering challenge. The goal of the current work was to show that, keeping these image-level features constant, graph processing improves network performance, and we expect this overall finding to hold with improved node features. In addition, while the same model architecture generalized across both animated and natural videos, we did not test a single trained model's ability to generalize, due to the different interaction types in the two video datasets. As a result, there is currently no direct evidence that transfer learning will work in SocialGNN. Further, unlike the Inverse Planning model, SocialGNN (like the other neural models tested here) was trained on human data. However, the model receives relatively little training data in each experiment. We believe with the right datasets, transfer learning across very visually different scenarios may work well, particularly if the right type of context information is added for each dataset. Transfer learning and optimizing context variables for different settings are interesting areas for future research that could further improve SocialGNN's ability to generalize.

In the past, the hypothesis that humans use only bottom-up, visual information to recognize social interactions has been dismissed due to the poor performance of purely visual models[4,19]. A large body of work has suggested that even infants evaluate social interactions based, not

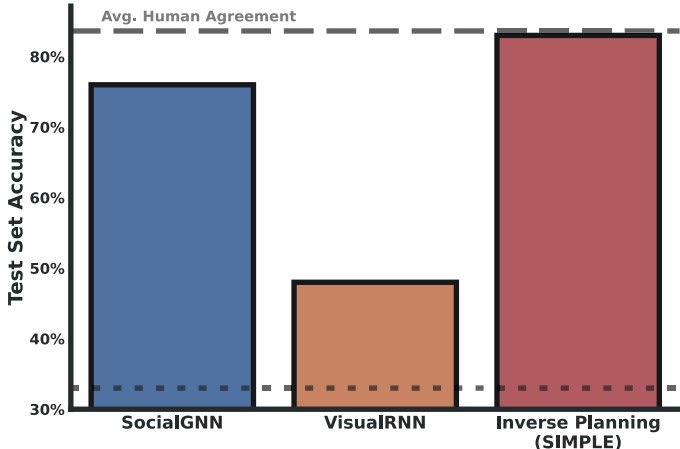

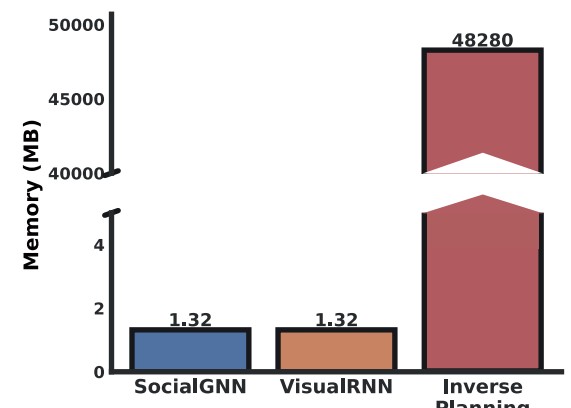

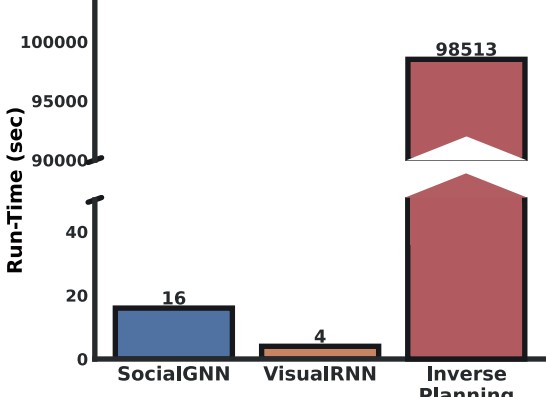

**Fig. 5 | Performance on the PHASE generalization set. a** Prediction Accuracy for the type of social interaction (friendly vs. neutral vs. adversarial) between two agents in the PHASE generalization set (100 videos). **b** Computational resources (run-time and memory) consumed by each of these computational models to make predictions on a subset of 25 videos.

on visual processes, but an innate moral code[9], which can be modeled via a generative framework[6]. The prior visual and cue-based models tested though have been overly simplistic focusing only on simple shape and motion features. Indeed, recent work has shown that infants' moral evaluations of social interaction scenes can be modeled via an associative learning mechanism linking agents and actions in a connectionist (bottom-up) framework[31]. However, it was previously unknown, how this information could be computed directly from visual input. These initial bottom-up, visual judgements may then be refined and supplemented by higher-level cognitive processing to give rise to the full range of humans' rich social scene understanding.

We find that a model with only visuospatial inputs can match human social interaction judgments when relevant inductive biases - here in the form of relational graph structure - are introduced. SocialGNN's graph structure constrains the relationship between entities in the scene, and thus allows the learning algorithm to prioritize one solution over others based on this knowledge. This is in line with developmental work suggesting humans have inherited useful inductive biases in the form of relevant data representations for flexible and efficient perception and cognition[32,33]. Perceptual biases, which focus a learner's attention on what information to prioritize for learning, could also be sufficient to produce the range of rich social behaviors we see in human infants. Different perceptual processes may have different kinds of inductive biases to structure incoming visual information[34]. For example, developmentally relevant object motion events have been shown to help computer vision systems learn about

hands and gaze direction[35]. Here we present a model with inductive biases for social scene understanding. These results also have implications for AI systems, as an added human-like inductive bias allows SocialGNN to make more human-like social judgments without incurring the computational cost of existing models.

SocialGNN is inspired by a growing body of recent work in graph neural network modeling for social behavior and multi-agent systems[4,36,37]. Unlike SocialGNN, however, these models all seek to predict agent trajectories, either as their final output or as an intermediate step towards a final social prediction. In contrast, SocialGNN does not predict agent trajectories and instead uses its graph structure to directly predict the relationship between agents, formalizing a crucial insight from cognitive science and development that humans view the world in the form of objects and relations[24,38]. Two other models[29,39] like SocialGNN operate directly on graphs of scene entities, but they either don't preserve temporal information or require intermediate supervision at smaller time scales. As a result, they are not well suited to match human behavior on extended events like the ones tested here[26].

Structured relational information, specifically in the form of graph representations, may be crucial to human social interaction judgments as we see that simply giving models relational information (as in the case of VisualRNN-Rel) is not enough to reproduce human behavior. Interestingly, the results of our ablation study further show that it is not sufficient to just represent the relationship between two social agents, but it is also important to represent the relationship of these

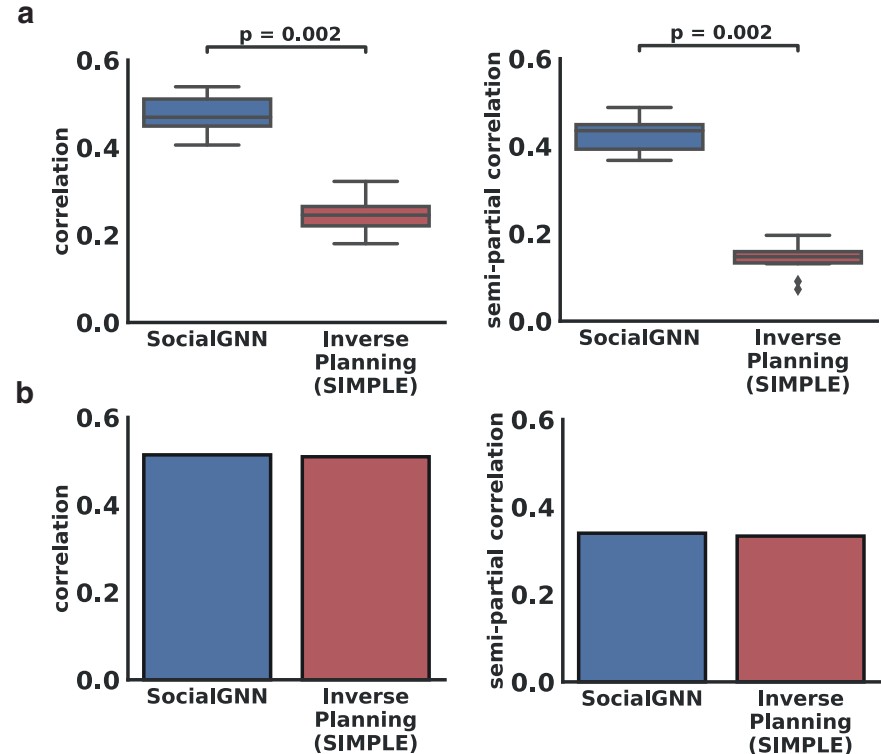

**Fig. 6 | RSA results on the PHASE datasets. a** Correlation and Semi-partial correlation between SocialGNN and the inverse planning model (SIMPLE) and human judgements RDMs for held-out test videos in the PHASE standard set. Box whisker plots showing each model's correlation (left) and semi-partial correlation (right) with human judgements across bootstraps. The center lines depict median, boxes middle 50%, diamond points the outliers, and whiskers accuracies outside the middle 50%. Significant differences are denoted by *p*-values (paired permutation test, *n* = 10,000 permutations, all two-tailed, Holm-Bonferroni corrected). **b** Correlation and Semi-partial correlation between SocialGNN and SIMPLE and human judgements RDMs for the PHASE generalization set.

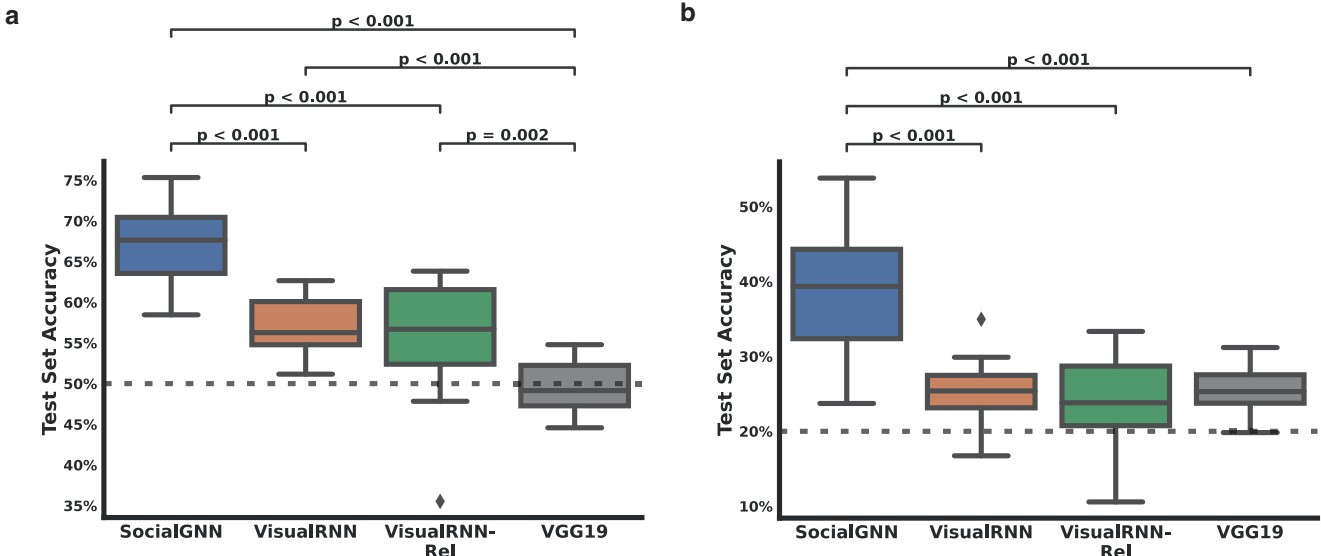

**Fig. 7 | Performance on the Gaze dataset. a** Prediction Accuracy for the presence of social interaction between two agents in the Gaze dataset. The chance accuracy here is 50% (dotted line). **b** Prediction Accuracy for the type of social interaction ("non-communicative" vs "mutual gaze" vs "gaze aversion" vs "gaze following" vs "joint attention") between two agents in the Gaze dataset. The chance accuracy here is at 20% (dotted line). Significant differences are denoted by *p*-values (paired permutation test, *n* = 10,000 permutations, all two-tailed, Holm-Bonferroni corrected). All other plotting conventions follow those in Fig. 4.

agents with other non-social entities (objects) in the scene to understand their social interactions. This suggests that representations of how agents interact with their physical world is necessary to gain a full social understanding of a scene. A major advantage of SocialGNN is the

flexibility it provides in representations and computations over all possible relationships in a scene, as the graph input determines how the entities interact. Further, it uses the same set of functions across all nodes and all edges, which may make the model capable of

combinatorial generalization in a human-like manner[25]. SocialGNN allows us to formalize and make a strong case for the theory that humans use graph-like visual representations to extract social information from scenes.

There are many aspects of recognizing and understanding social interactions that go beyond vision and rely on mental models and simulations of other agents' thoughts, beliefs, and actions. For example, understanding why two people are fighting or interpreting the goal of someone else's social actions. Integrative and dual-process theories of social cognition propose that these slower, cognitively laborious and more flexible processes (Type 2 processes), are separate from fast, efficient, and stimulus-driven perceptual processes (Type 1 processes), and both types account for different aspects of social cognition[40]. A growing body of behavioral[12–16], neuroscience[13,17,18], and now computational work, suggests that social interactions, and observations of them, are a fundamental part of the human experience that rely on visual processing. This work provides critical proof of concept that social interactions can be recognized based only on visuospatial information, and insight into the computations and representations that underlie this ability. It remains an open question the extent to which adults rely on visual processing versus mental simulation to interpret different social scenes, and whether this differs in infants and non-human primates. SocialGNN and the results presented here provide a computational framework to answer these questions and understand the neural and behavioral representations of social interaction scene understanding.

## Methods

All studies detailed here received ethical approval from the Johns Hopkins Homewood Institutional Review Board and complied with all relevant ethical regulations.

### PHASE dataset

The PHASE dataset consists of animated shape videos generated via a physical simulator and hierarchical planner, where two agents are moving around in a simple 2D environment, resembling real-life social interactions. There are also two objects in each video that the agents can carry/push. The environment has four landmarks and some walls, all stationary. The agents and objects (entities) can move over landmarks but cannot pass through the walls. The agents each have a physical or a social goal that is input to the video generator, and both are given eyes to lend more animacy to them. The sizes of the agents and objects, the strength of the agents, and the layout of the walls can all vary across the videos. The dataset includes a 400-video standard dataset and a 100-video generalization set with novel environment layouts and agents with social/physical goals that are unseen in the standard 400-video set. For training and testing within the standard set, we ran ten bootstrapped train/test splits of 300/100 videos. For the generalization set, we trained the model using the entire 400-video standard set and tested it on the 100-video generalization set. Each video is 5–25 s long. Some videos featured long time windows at the end with no movement, so we trimmed all the videos to end 2secs after all the entities stopped moving, resulting in 2.75–25 s long videos. For more details on the dataset see Netanyahu et al., 2021.

### Human behavior experiments

To compare human social evaluations on the PHASE dataset to different models, we collected human judgments of the social relationships between the two agents in each video, using the Prolific (https://www.prolific.com/) online platform. Informed consent was obtained from all participants before the experiment. Participants were given the following instructions: You will watch some videos with creatures moving around in a simple environment. There are two objects that the creatures can carry/push. There are four landmarks and some walls, all stationary. The creatures and objects can move over landmarks but

cannot pass through walls. After each video, you will be asked to describe the creatures' relationship with each other. The relationship could be Friendly/Cooperative, Neutral, or Adversarial/Competitive. (Fig. 1, Supplementary Fig. 1).

Participants were not told that these agents were pursuing specific goals. Following the instructions, participants were shown one example each of a typical "friendly", "neutral", and "adversarial" interaction. In each trial, the participants were asked to rate the relationship between the two creatures. Each participant rated 23 randomly ordered videos, including a random subset of 20 videos from the dataset and the three example videos that were shown in the instruction phase of the study as catch trials. Participants who did not complete the experiment or rate all three catch trials correctly were excluded from further analysis (77 participants excluded from the standard-set experiment, and 36 from the generalization-set experiment). We had a total of 318 participants in the standard-set (ages: 18–68, mean age = 28; sex: 157 Female, 148 Male, 13 Unspecified) and 103 participants in the generalization-set (ages: 19–72, mean age = 37; sex: 55 Female, 46 Male, 2 Unspecified). After exclusions, we were left with 241 participants for the standard dataset and 67 participants for the generalization set, with 10–15 ratings per video for the standard set (median number of ratings = 11.5), and 10–19 ratings per video for the generalization set (median number of ratings = 10). The sample size for this experiment was chosen such that there were at least 10 ratings per video, based on prior work with the same dataset[26]. Sex, gender, age, and ethnicity were not considered in the experimental design, and this information was self-reported before the experiment. The participants were compensated at a rate of $10 per hour and the median time to complete the experiment was 15 min.

We used the mode of the human ratings as the ground truth for model training and evaluation ($Y_{video\_i} = \text{mode}(\mathbf{R}_{video\_i})$). We also calculated the overall human agreement (HA) for each video as the ratio of the ratings equal to the mode of the ratings ($HA_{video\_i} = |\mathbf{R}_{video\_i} == Y_{video\_i}| / |\mathbf{R}_{video\_i}|$). Then we averaged across videos to get the overall human agreement for the train set and test set ($\mu(HA_{video\_i})$). This was done to get an estimate of the noise ceiling that takes into account the ambiguity in judging relationships, which therefore also existed in the models' training data.

### Human Gaze Communication Dataset (Gaze)

We used the VACATION (Video gAze CommunicATION) dataset consisting of 299 videos from YouTube with people interacting with each other and with objects in dynamic, natural contexts. Videos in the original dataset ranged in length from 2 to 64 s (56 to 1863 frames, median 260 frames), and contained multiple different gaze events. The gaze communication labels provided include "non-communicative", "mutual gaze", "gaze aversion", "gaze following", and "joint attention". Labels were collected by two human annotators in the original paper. As described in their methods, in videos where the two annotators disagreed a third specialist in the field assigned the label. We bootstrapped 20 train/test splits of 224/75 videos and then divided each video into shorter clips (median 50 frames) such that each clip only contained one type of gaze communication. Clips from the same video were kept together in either the train or test set to avoid having visually similar video clips across the two sets. To facilitate graph creation and help standardize graph size, we only kept clips with at least 2 people and a maximum of 5 entities (people + objects). After removing clips with anomalies (missing or inconsistent labels, e.g., single agent with "mutual gaze" label), we had ~740 video clips for training and ~215 clips for testing in each bootstrap. See Fan et al., 2019 for more details on the original dataset.

### SocialGNN

**Graph creation.** For each video we created a graph ($\mathbf{G}_{in}^t$) at each time step/frame (Fig. 2). The nodes ($\mathbf{V}_{inp}^t$) in the graph were the entities

(agents/objects) in the frame, the node features were various visuospatial properties of the entities, and the edges represented a relationship between the entities. For the PHASE dataset, node features included 2D position, 2D velocity, angle, size, and whether that entity was an agent or object. A bidirectional/undirected edge ($E_{inp}^t$) was added between two entities if they were in physical contact. Each graph also gets contextual information in the form of walls and landmarks coordinates.

For the Gaze dataset, node features were obtained by passing pixel information within that entity's bounding box through a pretrained VGG19 network[30]. The output from the penultimate fully connected layer was reduced to 20 dimensions via PCA and this feature vector was appended with the 4D coordinates of the bounding box (representing the location and size of the entity) and a boolean variable denoting whether it was an agent (person) or an object. The edges ($E_{inp}^t$) in these graphs were directed and are determined based on the labeled gaze direction of each agent. While gaze and bounding box information was provided in the Gaze dataset, this information is extractable via a range of bottom-up computer vision algorithms. For both datasets, the edges were binary and had no features.

**Model architecture.** At each time step, the graph representation of the video frame ($G_{in}^t$: $V_{inp}^t$, $E_{inp}^t$) is passed to a module called Gspatial, which learns an updated edge representation that takes into account the graph structure and features of all nodes. Specifically, the Gspatial module consists of a function $\Phi^e$: a linear layer that takes in the concatenated node features from nodes on each side of an edge, along with context information (Supplementary Fig. 3a), and outputs an updated representation for that edge. Updated representations for all the edges at a given time step are concatenated ($G_{spatial}^t$) and then passed to the Gtemporal module where a Long short-term memory (LSTM) unit (function $\Phi^s$) combines the current representation ($G_{spatial}^t$) with the information from previous time steps ($G_{state}^{t-1}$). This is repeated at each time step in a recurrent neural network (RNN)-like structure. The representation at the final time step ($T$) is passed through a linear classifier that is trained to predict the social interaction ($Y$) (e.g., friendly, neutral, or adversarial) (Fig. 3a and Fig. 3b). (Refer to Supplementary Methods: Experimental Settings for detailed parameter settings).

Due to the nature of the gaze dataset labels (provided for each agent), we evaluated a modified version of SocialGNN that makes predictions based on learned node, rather than edge, representations (Supplementary Figs. 2, 3b). The model operated the same as above, but with an additional process at each time step. Specifically, after updating edge representations as described above, node representations are updated using a Linear layer (function $\Phi^v$) that takes in, for each node, a concatenation of its node features and the sum of updated edge representations for all the edges that node is a part of (function $\rho^{e \rightarrow v}$), and outputs an updated node representation. After this linear layer, updated representations for all the nodes are concatenated ($G_{spatial}^t$) and then passed on to the Gtemporal module (Supplementary Fig. 3). This model provided similar performance to the unmodified SocialGNN (reported in the main text) on the PHASE dataset (Supplementary Fig. 4).

**Benchmark models**

**VisualRNN.** We compared SocialGNN to a baseline visual model with the same broad RNN architecture and input, but without the graph structure, called VisualRNN (Fig. 3c). This model gets the same input information as SocialGNN, but instead of the graph input and processing the LSTM takes in concatenated features of all the entities in the scene ($F_{in}^t$). We also compared SocialGNN with a modified version of this model: VisualRNN-Rel, where along with the entity features, a boolean vector denoting which edges are present is appended, to match the binary edge information provided to SocialGNN. For the

PHASE videos, to provide context information to the model, we followed the same procedure described for SocialGNN: we appended 2D coordinates of the 3 walls and 4 landmarks to the concatenated features ($F_{in}^t$) before passing it to the LSTM.

Trainable parameters, such as learning rate, regularization parameter, and sizes of all Linear/MLP/LSTM layers, are set to be similar across SocialGNN and matched visual models, and the hyperparameters for each model are tuned using 5-fold stratified cross-validation (see Supplementary Methods: Experimental Settings for detailed parameter settings).

**Inverse planning.** We also compared our model's performance on the animated videos (PHASE) with a generative inverse planning model called SIMPLE[26]. For each video, SIMPLE generates a hypothesis for the possible physical/social goals and relationships of the agents, simulates trajectories corresponding to these hypotheses, and compares these simulated trajectories to the observed trajectories of the entities in the video. The model selects the relationship with the best match between simulated and observed trajectories. To reduce its search space, SIMPLE first shortlists a set of hypotheses using a trained bottom-up model. These hypotheses are then updated over multiple iterations of trajectory simulation. For trajectory simulation, SIMPLE uses a hierarchical planner for each agent that generates subgoals and actions for a hypothesis. These actions are then executed via the physics engine to get the simulated trajectory. For more details refer to the model paper[26]. For comparison on the standard PHASE dataset, we got the predictions for all videos (except one video among the 400 where the model gave a NaN value) and calculated the performance accuracy for each bootstrap.

**VGG19.** We also compared our SocialGNN model with a standard visual CNN on the natural videos (Gaze dataset). We took a pretrained VGG19 (pretrained on ImageNet[30]). Although this model was trained to recognize objects, it has been shown that its learned representations generalize to different high-level visual tasks with fine-tuning[41,42]. We selected a model that operates on images (in our case frames of each video) rather than videos, since CNNs that operate on dynamic input have a large number of trainable parameters (requiring large training sets) and have been shown to have worse cross-task generalization[43]. We input pixel information from each entire frame of each video to the model. The output from the penultimate fully connected layer is then taken and reduced to 1500 dimensions using PCA. We averaged these features across frames for each video clip and passed it through a linear layer that we train to either do the 2-way or the 5-way classification on the Gaze dataset using the same cross-validation procedure as SocialGNN and the RNN models. While we do not have the scale of dataset to relearn all the model weights via backpropagation, our procedure is equivalent to fine-tuning the model on our tasks. See Supplementary Methods: Experimental Settings for trainable parameter settings such as learning rate, regularization parameter and class weights.

**Representational similarity analysis**

We conducted a Representational Similarity Analysis on the PHASE datasets to do an item-wise comparison between the two top performing models: SocialGNN and the Inverse Planning model (SIMPLE). We first created Representational Dissimilarity Matrices (RDMs) for both the models. For SocialGNN, we took the output of the final RNN step as the representation of a video, and for SIMPLE, we took the predicted probabilities for each hypothesis (friendly/neutral/adversarial) as the representation. Then, for each model, we calculated one minus the Pearson correlation between the representations of each pair of videos to give us the RDM for that model (using rsatoolbox[44]). Similarly, we created an RDM for the human ratings on these videos using the counts of "friendly", "neutral", and "adversarial" ratings given

to that video by all participants and standardize this array to sum to 1. For example, if a video was rated friendly by 8 participants, adversarial by 2, and neutral by 0, then the human ratings representation for that video would be (0.8, 0, 0.2). This representation captures the ambiguity in human ratings. Using these, we again calculate one minus the Pearson correlation between the representations of each pair of videos to give us the Human Ratings RDM.

We created these three RDMs for test videos in each bootstrap in the standard set, as well as the videos in the generalization set. To calculate the similarity between the computational models and human judgements we calculated the Spearman correlation between each model's RDM and the Human Ratings RDM. For the standard set, we reported the mean correlation across bootstraps. To measure unique variance explained by each model, we calculated the semi-partial Spearman correlation between each model RDM and Human Ratings RDM while controlling for the other model.

### Statistical inference
To test the significance of our model comparisons, we used scipy.stats's non-parametric paired permutation testing function with 10,000 resamples (shuffling within each bootstrap). We report Holm-Bonferroni corrected two-sided $p$-values. The $p$-values for comparisons on the PHASE standard set, are calculated using 1024 permutations instead of 10,000, which is the total number of distinct paired permutations possible with ten bootstrapped values.

In the RSA, for calculating the significance of the correlations and semi-partial correlations, we use permutation testing with 10,000 permutations of the video label. We again report two-sided $p$-values. For $p$-values for the correlations/semi-partial correlations on the PHASE standard set, we averaged across bootstraps to get the null distribution and compared it to the true mean across bootstraps.

### Estimating computational resources consumed
We get the amount of computational resources consumed by each model for prediction, using the "CPU Utilized" and "Memory Utilized" reported by the Slurm workload manager on the JHU Rockfish cluster.

### Reporting summary
Further information on research design is available in the Nature Portfolio Reporting Summary linked to this article.

## Data availability
Human behavior data included in this paper, along with the processed annotations from the PHASE[26] and Gaze[29] datasets are available on Github (https://github.com/Isik-lab/SocialGNN) and Zenodo (https://doi.org/10.5281/zenodo.8433260)[45]. The original videos and annotations can be downloaded from https://tshu.io/PHASE for the PHASE dataset, and requested from https://github.com/LifengFan/Human-Gaze-Communication for the Gaze dataset. Both raw and processed files have been provided for the human behavior data. Source data for all figures are provided with the paper. Source data are provided with this paper.

## Code availability
The analysis code and instructions to use the code are available on Github (https://github.com/Isik-lab/SocialGNN) and Zenodo (https://doi.org/10.5281/zenodo.8433260)[45].

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

## Acknowledgements

We would like to thank Mick Bonner, Alan Yuille, Josh Tenenbaum, Andrea Tacchetti, and Tianmin Shu for useful discussions of this work, as well as Tianmin Shu and Aviv Netanyahu for sharing and assisting with the PHASE dataset, Lifeng Fan for sharing the Gaze dataset, and Jihoon Kim for help preprocessing the Gaze Dataset. We would also like to thank Emalie McMahon, Haemy Lee Masson, Mick Bonner, and Shari Liu for feedback on the manuscript. This work was funded by NIMH R01MH132826 awarded to L.I.

## Author contributions

M.M. and L.I. developed the concept and designed the experiments; M.M. designed and implemented the computational models; M.M. analyzed the results, which were then interpreted by both M.M. and L.I.; M.M. and L.I. wrote the manuscript.

## Competing interests

The authors declare no competing interests.
