## [Peer Review File · Nature Communications]

Reviewers' Comments:

Reviewer #1:

Remarks to the Author:

The current manuscript established a SocialGNN model to predict human judgments of others' social interactions, based on relational visual information. The model outperformed the VisualRNN model, VisualRNN-Rel model and inverse planning model in animated shape videos, PHASE dataset. The authors then extended the SocialGNN model to a Human Gaze Communication dataset and verified the model performed well in predicting human responses in natural videos with gaze information. The authors concluded that humans can successfully make social interaction judgements based on relational visual information. It is quite interesting and innovative to use the idea of the graph neural network to build a computational model to classify social interaction. The study demonstrates the power and advantage of the graph neural network. However, I cannot recommend the paper for publication at this stage given the following concerns.

First, this manuscript concluded that recognizing human social interactions can be accomplished only using visual information. This conclusion is too hasty. There is no doubt that relational visual information plays an important role in classifying social interaction, but other information is also not negligible. The contribution of each factor greatly depends on the scenarios. In PHASE dataset, stimuli are geometric shapes. In such scenarios, relational visual information is an important cue that humans can rely on, thus, it is quite reasonable that SocialGNN is effective in these datasets. However, referring to Human Gaze Communication dataset which is more natural, the classification accuracy of the SocialGNN test appears to be relatively low, less than 70% in judging interaction status and around 40% in judging interaction types. The result indicates that considering only relational visual information is not enough for these natural scenes.

Moreover, even for visual information, the study oversimplifies visual information. Visual information included many aspects, such as the distance between agents, the facing orientation of agents, and gestures and expressions of the agents. These cues could contribute to judging the state of social interactions, depending on the information available in a specific scenario.

Another concern is the generalization of the model across different datasets. If I understand correctly, for each dataset, the authors divided the dataset into training and testing sets, trained neural networks by the training set, and predicted data in the test set. Although the networks trained in different types of datasets are named SocialRNN, they were actually different networks rather than the same one. In Fig 5, the authors used 100 new videos as the test set to prove the generalization of their models, these 100 videos were still the same type of stimulus as PHASE dataset. From an engineering application perspective, it is important to develop a model that can generalize to different types of datasets. It would be interesting to know whether the model trained by the data in PHASE dataset can predict the data in Human Gaze Communication dataset or vice versa.

The third concerns are the benchmark models used in the study are weak and look like "strawmen." Defeating very weak component models does not prove that SocialINN is a good model. The decoding accuracy of VisualRNN and VisualRNN-Rel were close and even lower than the chance level for Human Gaze Communication. The confusion matrices in Fig. S7 showed that VisualRNN judged all samples of the Gaze dataset as interactive groups and VisualRNN-Rel demonstrate the same trend. In Fig S8, the output of VisualRNN and VisualRNN-Rel were either Avert Gaze or Joint Attention. None of the examples was judged as the rest three types: Gaze Follow, Mutual Gaze or Single Gaze. The classification results of VisualRNN and VisualRNN-Rel were so biased that raised concerns about whether the two models are fair benchmark models for comparison.

Other points

1. For PHASE dataset, there is a considerable inconsistency between Human Ratings and PHASE Labels (Table 1), and the degree of agreement between human participants was only 80%. It is questionable whether PHASE dataset is not a good choice for use to train a neural network. Would it be possible to exclude the controversial videos and leave ones with a high agreement between

participants?

2. In the test accuracy calculation, it is not very proper to directly use the chance level as the baseline since machine learning algorithms are sometimes very powerful at getting results beyond the chance level even for some random data. Therefore, the shuffled baseline should be calculated. For example, labels of all the data in the dataset should be randomly shuffled and the same model with the same training process should apply the shuffled data. The process could be repeated thousands of times to estimate the shuffled baseline.

3. I understand the authors used bootstrapped train/test splits. What is the number of iterations during bootstrapped?

4. Compared with Figure 4, the predicted accuracy of the inverse planning model in Figure 5 had a great improvement, reaching nearly 90% in Figure 5. As Figure 5 uses 100 new videos as the test set, which is a more difficult task. What is the reason the inverse planning model performed better under such conditions?

5. Referring to Human Gaze Communication Dataset, why not ask human participants to rate these stimulus videos as human participants did for PHASE dataset?

6. Confusion matrices of SocialGNN, VisualCNN and Inverse planning model did not report for PHASE dataset.

7. It is a bit surprising that the authors no longer compared SocialGNN with the inverse planning model in Human Gaze Communication dataset. Is there any particular reason for that?

8. The authors did not provide a run-time calculation for Human Gaze Communication dataset. It would be interesting to see this analysis.

9. For figures with multiple subplots such as Figures 3, 5, and 6, I recommend labelling each subplot with Letters A B C to improve the readability.

10. 46 lines are missing a period "recognizing social interactions Social interactions".

Reviewer #2:

Remarks to the Author:

Summary and overall evaluation:

This paper presents a new computational model, SocialGNN, which captures how humans rely on relational visual information to recognize social interactions. I am not an expert on graph neural networks, so I cannot comment on the low-level implementation choices. But, if everything was executed correctly (and from what I can tell, it was), this paper is a breakthrough in both cognitive science and artificial intelligence.

In cognitive science, this is (to my knowledge) the first computational model of social computations in high-level vision. This is an incredibly important gap that needed to be filled. To date, we have models of high-level social cognition (e.g., meta-representational belief reasoning), and of low-level social perception (e.g., face detection and pose estimation), but nothing in between. As consequence we understand little about the intermediate computations that interface low-level representations of agents with the machinery of high-level cognition. This paper fills this gap, and in doing so it opens the door to studying the full social reasoning pipeline computationally. This work has the promise of revolutionizing social neuroscience, in a similar way to how deep neural network models of object perception have revolutionized how we study the ventral stream.

This paper is also a great contribution to artificial intelligence. Despite the everyday breakthroughs in AI, these advances are usually limited to categorization, planning, and language, with surprisingly fewer advances in human-like social intelligence.

In short, I am very enthusiastic about this paper, and think it is a rare and valuable contribution.

Main areas requiring revisions:

I was surprised that the very limited space in the introduction was used to argue that inverse planning was wrong. I have no doubt that it is in many ways, but I did not see how this paper speaks to that issue. I do not think that the paper currently speaks to that and I wonder whether

the conflict being set up in the introduction is an artificial one (and, as I see it, SocialGNN is a clear stand-alone contribution).

I believe this stems from a few related points:

1. Inverse planning is explicitly a framework for modeling high-level meta-representational Theory of Mind. The introduction frames this paper around the idea that inverse planning cannot be a model of visual perception, but I do not think that this a position that anyone holds.

The paper justifies this position by saying that people have used inverse planning to model how we infer social relations, but social relations are detected in high-level vision. The issue here is that inferences about social relations happen in both high-level vision and in high-level cognition. So, we need models of the inferences that happen at both levels, and I do not follow why the paper has the implicit assumption of that the inferences must either happen exclusively in high-level vision or exclusively in high-level cognition (and therefore, one of the models must be wrong).

I agree that the early modeling how ToM helps infer social relations used experimental paradigms that were too simple and that can be solved by vision alone (e.g., Ullman et al 2010). Therefore that work moved away from the simple helping/hindering paradigms to include false-belief reasoning. If you consider, for instance, Hamlin et al., 2013, those tasks are clearly not solved by vision (see in particular the complex false belief conditions).

This makes me wonder whether the presented conflict between IP and SocialGNN is artificial. It might be more accurate to say that we have models of meta-representational Theory of Mind, but that there are a lot of social computations happening in high-level vision that we do not yet understand and that is the goal of this paper. The comparison against inverse planning is still very useful, but it is not obvious to me that the framing on "only one can be right" makes sense.

If there are papers that propose inverse planning as a model of visual perception, or argue that all inferences about social relations are driven by inverse planning, it would be helpful if this were cited in the introduction. In that case, it would be helpful to explain this as a position held by some people (since I think this is a fringe position), rather than an intrinsic component of inverse planning.

2. My understanding is that Inverse Planning is a proposal at Marr's computational level of analysis and is currently agnostic about the underlying inference algorithm. While some implementations use sampling, this is a computational placeholder for whatever inference mechanism might be implemented in the brain. For instance, one of the central papers proposing inverse planning (Baker et al., 2017) explicitly argues that compiling these inferences into a neural network architecture might be better for capturing the algorithmic level (like, e.g., Yildirim, et al., 2020).

As analogy, consider the proposal of perception as inverse graphics. Initial support for this idea was found through a first generation of models that aimed to test the value of the theory at a computational level of analysis, and then underwent a second generation implemented in a neurally-plausible way. The second generation of models did not imply that inverse graphics was wrong, and it would have been inaccurate to say that inverse graphics was intrinsically committed to top-down sampling just because this was a computational approximation of testing if the outputs of the two models matched.

In the context of this paper, some statements, like lines 36-39, are inaccurate. I don't think that the heart of the framework is to compare trajectories to internal simulations. Rather, it's the idea that the inference mechanism is approximating the inversion of a planner.

This makes the claims about run-time and memory load of inverse planning feel misleading. That being said, the memory load comparisons are still super important for discussing the value of SocialGNN in the context of AI. The paper just needs to be clearer about what's at stake in each claim.

3. With the two points above in mind, for the paper to deliver on the promise of the introduction, it

would need to show that inverse planning is wrong, which requires two pieces of evidence that aren't in the paper yet.

First, you would need to show that SocialGNN is not a neural implementation of inverse planning.

For instance, you could show that SIMPLE and SocialGNN are uncorrelated, and that SocialGNN is more human-like on events where the models show the sharpest disagreement. I think this will be true based on the accuracy results shown in Figure 4, but it's still possible that this accuracy difference arises from relative confidence differences, and not qualitatively different computations. This argument could also be bolstered by analyzing the pattern of errors more clearly (more about this on point X below).

Although this isn't strictly necessary for publication, this would be a valuable addition. There is the caveat that I don't think anyone believes that inverse planning is a model of visual perception (more on this below).

Second, you would need to show that SocialGNN is better than inverse planning in the domains that inverse planning is proposed to work. For this, you would need to have false-belief style social inference tasks. My read is that the authors do not actually believe that this is the case. I found in particular the discussion to have much more nuance that was missing in the rest of the paper.

A few more areas where clarity would be helpful:

4. In the introduction the paper claims that inverse planning uses explicit representations of other agents' minds and that this conflicts with evidence that vision recognizes social interactions (lines 43-46). I did not follow this argument.

Is it not possible for social interaction to trigger explicit representations of minds within vision? It seems to me that a social representation of helping or hindering is, by definition, an explicit representation of a goal.

5. The discussion also states that SocialGNN does not have explicit goal representations, and that this implies that visual information alone is enough for humans to recognize social interactions (lines 296-300). I also did not follow this argument, for two reasons.

First, how do we know that SocialGNN is not explicitly representing goals? To show that, don't you need to show that there is no information in SocialGNN that exclusively represents goals?

Second, even if SocialGNN does not have explicit representation of goals, why does that imply that visual information alone is enough to recognize social interactions? Don't we already know that visual information alone is enough? Wouldn't this be true independent of whether the inferences happen in high-level cognition or not?

6. "In the past, the hypothesis that humans use only bottom-up, visual information to recognize social interactions has been dismissed due to the poor performance of purely visual models." (lines 308-309).

I was not aware of this. Are there any references that could be added?

7. I found it difficult to get a sense of what the models were doing based only on the abstract metrics. Is it possible to include a sub-set of stimuli/trials with the corresponding inferences from each model. It would be helpful to get a richer sense of what the models looked like in a more trial-by-trial basis.

8. The main text alludes to the fact that SocialGNN makes more human-like errors and directs readers to supplemental information. The text here is pretty vague and there is no proper analyses of the errors. To keep that claim, you might want to extend this into a proper and detailed statistical analysis.

Minor:

- Missing period in line 46.

- I could not understand the sentence in lines 56-58.

- Inverse Planning is a general framework, but the paper tests a very specific instantiation of it. That's fine, but a lot of the metrics reported report idiosyncrasies about a very specific implementation. It would be more accurate to use the term SIMPLE in the results and figures (the run-time and memory values reported in Fig. 5 are about SIMPLE, and not a general signature of inverse planning).

- Discussion on lines 309-312 might be inaccurate. It pits inferences in vision as conflicting with an innate moral code. Why can't it be both? I think the community arguing for an innate moral code would be perfectly happy with an account where vision detects first-order relations (friendly, adversarial, etc), which are then supplemented by high-level cognition (e.g., seemingly friendly but with bad intentions; seemingly adversarial but they're looking out for their friend from making a bad choice; etc).

- One other small point is that the paper often talks about 'inferences based on visual information' when I think it means to say 'inferences happening within the visual system'. I found this to be about confusing.

- While the paper is correct that inverse planning in its most general formulation is intractable, it actually does become tractable when situational constraints are added. See Introduction claims that inverse planning is intractable, but it isn't! See Blokpoel, Kwisthout, van der Weide, Wareham, & van Rooij (Journal of Mathematical Psychology).

Summary of suggested changes and final remarks:

This is a fantastic paper and all concerns listed above can be addressed. There are multiple types of revisions that would satisfactorily address these comments, but here are two possible routes:

1. The case against inverse planning.

The paper could be revised to make a tighter case against inverse planning and show stronger evidence against it. To achieve this, you would need to identify the special position of inverse planning as a model of high-level vision and show that SocialGNN is not a compiled version of inverse planning, but a fundamentally different type of computational process. Based on the presented results, I am very optimistic this would come out.

2. Filling the gap in modeling social intelligence.

The paper could be revised to highlight the gap that SocialGNN fills. We have models of mid-level vision, and of meta-representational social cognition, but there are these rich inferences performed in vision and we have no models for that, and here is where SocialGNN. Critically, this version should still include the inverse planning comparisons, but consider using it as a proxy to show that these are not high-level ToM inferences (rather than saying that only one model can be right). In this case, I think it would still be valuable and important to show that SocialGNN is not an implementation of inverse planning, although it is not critical for publication.

Again, this is impressive work!

Reviewer #3:

Remarks to the Author:

Review

Summary of paper: The paper proposes an algorithmic-level model based on graph neural networks (GNNs) to explain human social inferences. The GNN model (referred to as SocialGNN)

takes as input spatial and physical information about a scene (e.g. the positions of objects, or gaze directions), performs a single step of relational inference at each timestep, aggregates the relational inferences across time via an RNN, and then decodes the result into a social judgment. This model is compared to two alternative models (a non-relational RNN, and a Bayesian Inverse Planning model) across two tasks (the PHASE dataset, which is a 2D animated dataset of Heider-and-Simmel-like scenes; and the Human Gaze Communication dataset, which involves naturalistic videos with annotated gaze/person/object information). SocialGNN achieves higher test set accuracy than either the RNN baseline or Inverse Planning model, and also lower computational cost than the Inverse Planning model. The paper concludes that SocialGNN is a better model of human social inferences, and as such, lends support to the hypothesis that social inferences may be performed bottom-up via perception and do not always require planning/mental simulation.

Summary of review: This paper studies an important topic (explaining human social inferences) and investigates a promising class of models from machine learning. I think this is really important research to do, and it's fantastic to see work which tackles this! In particular, while existing models (like the Inverse Planning model) are great at explaining human judgments, they are not always algorithmically very plausible, are very much limited in their generalizability due to their computational burden, and can't be easily applied when symbolic representations aren't available. As such, identifying algorithmic-level models which can more efficiently and effectively implement computational-level goals is a pressing concern in computational modeling of social cognition. However, while I laud the paper's goals, I believe it falls somewhat short on a few dimensions regarding the model's architecture, analysis, and baselines (see detailed comments below). I believe this could be a great paper, but also that it needs a bit more work.

Detailed comments, roughly in order of importance:

The paper claims that SocialGNN "generalizes" to different settings, but to achieve this, three different variants of SocialGNN are required (small edge update model without context, larger edge update model with context, and node update model). It feels like a stretch to say that the same model is generalizing across these different settings because these are really two different models under the most generous interpretation, maybe three under a less generous interpretation. (It would be much more compelling if exactly the same architecture were used, especially in terms of using the same edge/node updates across the PHASE and Gaze datasets, and in terms of using the same context across the two splits of the PHASE dataset).

The analyses in the paper focus only on comparing average accuracy across models. This is a fairly weak measure of how well the models explain human cognition, as it does not capture any nuances of the behavior. For example, how well does the model explain individual scenes? Is it more unsure on scenes that have higher disagreement amongst humans? etc. (In analyzing the PHASE dataset, the variance across human judgments is discarded, which is a shame---this is rich information that would be really revealing when compared to the model. For example, for each scene you could compute the KL divergence between the distribution of human judgments and the distribution/logits produced by each model. The paper's claims would be much stronger if you could show that the model is uncertain on scenes in which people are uncertain, and more certain when people agree. I realize there is some analysis along these lines in Fig S5, but this just looks at whether the model is correct/incorrect as a function of human agreement with the model, which is not the same thing as comparing the distributions of the model and of people. It's also not obvious to me that the difference between the models in this figure is statistically meaningful.)

I believe the paper uses the term "visual" incorrectly. For the PHASE dataset, SocialGNN does not take visual information as input: it takes spatial information. Visual information would be, for example, pixel values. An object's location, velocity, size, etc. are not visual properties but are physical or spatiotemporal properties. This may seem like a nitpick, but I think it's an important distinction because inferring spatiotemporal properties from visual input is a nontrivial task and is still, in some cases, an unsolved problem in computer vision. Stating that SocialGNN can infer social information directly from visual input is therefore attributing it more ability than it actually has. (For the Gaze dataset, it's somewhat more accurate because the CNN takes visual information as input, though it is still given object locations and sizes in the form of bounding boxes, as well as

gaze directions, which are spatial/physical information). Here are some locations I noticed where the term "visual" is used incorrectly: in the title, the names "VisualRNN" and "VisualRNN-Rel", lines 64, 71, 119, 129, 195, 196, 221, 230, 296, 299, 319, 348, 359, 373, 444, 490, and 502.

The detailed results with the RNN baseline---in which it collapses into making constant predictions---suggest to me that there is a bug. My suspicion would be that the model size is not large enough to capture the information in the task. It's also not really the most compelling baseline, to be honest, because concatenating all the object information and feeding it through an LSTM is an atypical choice and one that I wouldn't guess would work well for *any* task. (A better baseline would be a CNN or Vision Transformer architecture). Regardless, it should at least be able to memorize the training data; if it cannot do that, then it is not expressive enough.

From a machine learning perspective, the SocialGNN architecture is rather atypical and I would be surprised if it works for more complex tasks. Perhaps that is fine for a cognitive model, but it might limit the generality of the model if it won't work in more complex settings. Could you please elaborate on how you chose the particular architecture, what alternatives you tried, and how they performed? In particular, why not use a more typical GNN architecture, e.g. a recurrent GCN or recurrent MPNN?

In the Gaze dataset, how are the labels determined? The paper remarks upon "the importance of using human ratings rather than the default labels" (line 108) for the PHASE dataset, but then seems to rely on the default labels for the Gaze dataset. Were these labels similarly determined from multiple human judgments? As it's currently presented, the results on this dataset don't appear particularly relevant to validating models of human cognition because I am not sure how much they're actually capturing about human judgments vs. just whatever the annotator said. At a minimum, it would be helpful to discuss how the labels were collected (e.g. how many judgments per label, what the population was, etc.). As discussed above, it would be even better if it were possible to perform some more fine-grained analyses (e.g. does the model's distribution match the distribution of human ratings on individual stimuli).

Lines 199-200: "SocialGNN even outperforms the Inverse planning model on the standard PHASE dataset." I'm not sure this is a meaningful comparison. SocialGNN is trained in a supervised way to predict human judgments, and tested on similar (in-distribution) scenes. Given an expressive enough NN model class, any NN trained on this data will be able to do well on in-distribution scenes. In contrast, the Inverse Planning model is not given the same training data and is able to explain human judgments instead given a few assumptions about planning/goals/etc. That doesn't invalidate the utility of a model like SocialGNN---especially when compared to other NN architectures trained on the same data, it suggests which architectural features contribute to sufficient expressivity---but it does feel a little bit like comparing apples to oranges to compare it to the Inverse Planning model in this way. (I think it's telling that when generalizing out of distribution, the Inverse Planning model does much better than SocialGNN).

How many model runs/seeds were used to compute the results?

Lines 335-338: "SocialGNN is inspired by a growing body of recent work in graph neural network modeling for social behavior and multi-agent systems (Shu et al., 2021; Sun et al., 2022; Tacchetti et al., 2018). Unlike SocialGNN, however, these models all seek to predict agent trajectories, either as their final output or as an intermediate step towards a final social prediction." If there are existing NN-based cognitive models of social predictions, I would think they should be included for comparison, even if they predict agent trajectories as an intermediate step: isn't one of the goals of the paper to demonstrate that such predictions aren't necessary? To validate this claim, comparisons should be included to similar NN models which do include such assumptions.

Lines 348-349: "giving standard visual models relational information (as in the case of VisualRNN-Rel) is not enough to reproduce human behavior". I think it's a stretch to call VisualRNN a "standard visual model". A standard visual model would be a CNN. The architecture of VisualRNN is actually quite unusual, as I discussed above.

Lines 356-358: "it uses the same set of functions across all nodes and all edges, making the model capable of combinatorial generalization in a human-like manner". This statement is unsupported; I don't think the paper has demonstrated that this model achieves combinatorial generalization.

Lines 411-412: "Participants who did not rate all three catch trials correctly were excluded from further analysis". How many people were excluded in total?

Lines 37-39: "These models infer social relationships by comparing the observed agent trajectories to internal simulations based on not only visual information, but also inferred physical information about the scene and hypothesized goals of the agents." To be fair, SocialGNN also provides inferred physical information about the scene. It also is given access to information about what different types of social interactions look like, in the form of its training data, which other models do not have access to (as discussed above).

Lines 56-58: this sentence is ungrammatical.

REVIEWER COMMENTS

Reviewer #1 (Remarks to the Author):

The current manuscript established a SocialGNN model to predict human judgments of others' social interactions, based on relational visual information. The model outperformed the VisualRNN model, VisualRNN-Rel model and inverse planning model in animated shape videos, PHASE dataset. The authors then extended the SocialGNN model to a Human Gaze Communication dataset and verified the model performed well in predicting human responses in natural videos with gaze information. The authors concluded that humans can successfully make social interaction judgements based on relational visual information. It is quite interesting and innovative to use the idea of the graph neural network to build a computational model to classify social interaction. The study demonstrates the power and advantage of the graph neural network. However, I cannot recommend the paper for publication at this stage given the following concerns.

First, this manuscript concluded that recognizing human social interactions can be accomplished only using visual information. This conclusion is too hasty. There is no doubt that relational visual information plays an important role in classifying social interaction, but other information is also not negligible. The contribution of each factor greatly depends on the scenarios. In PHASE dataset, stimuli are geometric shapes. In such scenarios, relational visual information is an important cue that humans can rely on, thus, it is quite reasonable that SocialGNN is effective in these datasets. However, referring to Human Gaze Communication dataset which is more natural, the classification accuracy of the SocialGNN test appears to be relatively low, less than 70% in judging interaction status and around 40% in judging interaction types. The result indicates that considering only relational visual information is not enough for these natural scenes.

Thank you for your overall comments and for raising this concern. You are right that our model does not fully explain human performance on natural videos, particularly for categorizing the individual interaction types, and we now address this in the discussion (lines 344-359):

We note though that while SocialGNN performs significantly above chance on both natural video experiments, it is still far from the level of human performance in this domain. It seems likely that better engineered node features (image-level information about the agents and objects in each video) would improve the performance of all visual models. Optimizing these features is clearly an important engineering challenge. The goal of the current work was to show that, keeping these image-level features constant, graph processing improves network performance, and we expect this overall finding to hold with improved node features.

We consider this work on natural videos to be largely proof of concept. There is still much engineering work to be done to improve performance, which we believe will involve optimizing node features (i.e., extracting better image-level features for the objects and agents in each scene).

The primary claim of the paper is that, holding these image-level features constant, graph processing significantly improves match to human behavior. Optimizing these node features is clearly an important engineering challenge but is not the goal of the current investigation.

Moreover, even for visual information, the study oversimplifies visual information. Visual information included many aspects, such as the distance between agents, the facing orientation of agents, and gestures and expressions of the agents. These cues could contribute to judging the state of social interactions, depending on the information available in a specific scenario.

While we did not extract the cues directly from pixelwise values, we only included input information to our model that could be extracted in a straightforward way from pixelwise information. The PHASE dataset is a very simplified visual stimulus so extracting properties like the shape, size, speed, and orientation of the entities would not be difficult (in fact many of these properties could be extracted based on object color alone). For the Gaze dataset, automatic computer vision systems currently do a good job of detecting bounding boxes and gaze direction (we have used the Amazon Rekognition platform for similar purposes, and there are also several classifiers for face and eye detection in OpenCV). While we believe it would be fairly straightforward to extract these properties in the current paper, we don't think it meaningfully adds to the intellectual contribution of the paper.

Importantly, we did not include gesture or expression (though perhaps some visual correlates of these are represented implicitly in the VGG activations for the natural videos experiment). If the addition of gesture or expression information to SocialGNN is implied somewhere specific in the text, we would appreciate a pointer so we can clarify.

More generally though, when we discuss visual information, we do not mean simple pixelwise cues, but instead information that can be extracted or represented entirely in the visual system (i.e., the distinction between perception vs. cognition). Lots of seemingly high-level social information is extracted perceptually (animacy, goal-directed behavior) here we present evidence that SI should be included in this list. We have clarified this point in the introduction lines 63-69:

Note here that by “visual” we do not mean that humans exploit very simple pixel-level information to recognize interactions. Nor, on the other hand, do we mean visual information that is processed by higher-level cognitive systems. Instead, the above evidence suggests that like other high-level features, including causality and animacy (cite Scholl & Tremoulet, 2000), social interactions are also processed within the visual system

using spatiotemporal cues in a manner that is distinct from cognitive processing (Firestone & Scholl 2016).

We agree though that terms like “visual information” might be misleading, so we have replaced these with “visuospatial information” or “information extracted by the visual system” and aimed to clarify the contrast with the cognitive system.

Another concern is the generalization of the model across different datasets. If I understand correctly, for each dataset, the authors divided the dataset into training and testing sets, trained neural networks by the training set, and predicted data in the test set. Although the networks trained in different types of datasets are named SocialRNN, they were actually different networks rather than the same one. In Fig 5, the authors used 100 new videos as the test set to prove the generalization of their models, these 100 videos were still the same type of stimulus as PHASE dataset. From an engineering application perspective, it is important to develop a model that can generalize to different types of datasets. It would be interesting to know whether the model trained by the data in PHASE dataset can predict the data in Human Gaze Communication dataset or vice versa.

We agree such transfer learning experiments between datasets would be exciting and compelling, but unfortunately, they are not feasible with the current datasets we have since the two datasets have fundamentally different types of interactions (Friendly, neutral adversarial versus different types of gaze). This type of experiment would therefore require creating a new shape and/or natural video dataset that are compatible to cross training, which we believe is ultimately beyond scope of this paper. We do think our current claims about generalization though are well founded and we have tried to strengthen them in revision:

1. The 100-video generalization set in PHASE were explicitly designed to include different visual scenes and social scenarios NOT included in the training set. We believe this is analogous to someone experience a new social situation in a new environment (but in the same world with the same type of social partners where the same laws of physics apply).
2. We have updated the model architectures for each experiment, so now the same architecture (trained on different tasks) is used in each experiment (updated figures 4-6, S4). In the results and discussion, we clarify that we are referring to generalization in model architecture.
3. Finally, we note that our training set in all experiments is relatively small by machine learning standards (300-400 videos) suggesting there is not much situation-specific of parameter tuning occurring and that with the right datasets cross-dataset generalization would likely be very feasible. We have added discussion of these

points to lines 351-359:

In addition, while the same model architecture generalized across both animated and natural videos, we did not test a single trained model's ability to generalize, due to the different interaction types in the two video datasets. However, the model receives relatively little training data in each experiment. We believe with the right datasets, transfer learning across very visually different scenarios would work well, particularly if the right type of context information is added for each dataset. Determining optimal context variables for different settings is an interesting area for future research that could further improve SocialGNN's ability to generalize.

The third concerns are the benchmark models used in the study are weak and look like "strawmen." Defeating very weak component models does not prove that SocialINN is a good model. The decoding accuracy of VisualRNN and VisualRNN-Rel were close and even lower than the chance level for Human Gaze Communication. The confusion matrices in Fig. S7 showed that VisualRNN judged all samples of the Gaze dataset as interactive groups and VisualRNN-Rel demonstrate the same trend. In Fig S8, the output of VisualRNN and VisualRNN-Rel were either Avert Gaze or Joint Attention. None of the examples was judged as the rest three types: Gaze Follow, Mutual Gaze or Single Gaze. The classification results of VisualRNN and VisualRNN-Rel were so biased that raised concerns about whether the two models are fair benchmark models for comparison.

For the PHASE dataset, the inverse planning model was introduced as the state-of-the-art benchmark. A very similar model to our VisualRNN (2-Level LSTM) was the next best model of relationship prediction so we believe the comparisons to be fair on this benchmark, and we now mention this when we introduce the model in the Results section (lines 184-185):

VisualRNN is an implementation of a Cue-Based LSTM, a standard perceptual baseline used in similar tasks (Netanyahu et al., 2021; Shu et al., 2020).

We take your point though that the VisualRNN models are not well suited to the natural videos in the gaze dataset. We find that to be a compelling part of our results. A small and well understood tweak (the addition of graph input and processing) to our model leads to a large improvement in accuracy.

To further address your concerns, we first performed additional fine-tuning on these models to optimize them for natural images. This led to improvements in their accuracy and the models are no longer outputting single class labels (see updated Figure 7). However, both models still perform significantly worse than SocialGNN.

We next tested a more standard CNN model typically used to recognize information in natural images, VGG-19 (see updated Figure 7). We include a justification for this model selection in Methods (lines 584-599)

We also compared our SocialGNN model with a standard visual CNN on the natural videos (Gaze dataset). We took a pretrained VGG19 (pretrained on ImageNet, (Simonyan & Zisserman, 2014)). Although this model was trained to recognize objects, it has been shown that its learned representations generalize to different high-level visual tasks with fine-tuning (Geirhos et al., 2021; Schrimpf et al., 2020). We selected a model that operates on images (in our case frames of each video) rather than videos, since CNNs that operate on dynamic input have a large number of trainable parameters (requiring large training sets) and have been shown to have worse cross-task generalization (Kataoka et al., 2020). We input pixel information from each entire frame of each video to the model. The output from the penultimate fully connected layer is then taken and reduced to 1500 dimensions using PCA. We averaged these features across frames for each video clip and passed it through a linear layer that we train to either do the 2-way or the 5-way classification on the Gaze dataset using the same cross-validation procedure as SocialGNN and the RNN models. While we do not have the scale of dataset to relearn all the model weights via backpropagation, our procedure is equivalent to fine-tuning the model on our tasks. See Supplementary Section: Experimental Settings for trainable parameter settings such as learning rate, regularization parameter and class weights.

Despite its success in matching human behavior across a range of other domains, it does a poor job of matching human social interaction judgements. While there are more advanced CNNs available, they in general don't do a much better job of explaining human behavior across high-level visual tasks (e.g., Geirhos 2021, Schrimpf 2018). While it is possible that with enough data end-to-end training of a vision transformer or CNN would do a better job at this task, unfortunately we don't have access to a labeled dataset at that scale. We also think that the relative success of SocialGNN highlights the efficiency of using a relational inductive bias in these low-training data settings.

Other points

1. For PHASE dataset, there is a considerable inconsistency between Human Ratings and PHASE Labels (Table 1), and the degree of agreement between human participants was only 80%. It is questionable whether PHASE dataset is not a good choice for use to train a neural network. Would it be possible to exclude the controversial videos and leave ones with a high agreement between participants?

Thank you for this suggestion. We agree that simply trying to match the median human response in ambiguous videos may not be a good strategy. On the other hand, we believe capturing ambiguity is important for models of human cognition. In this way we think the PHASE videos are well suited to this task.

To address this, we have now included representational similarity analysis (RSA) between human behavior and the models to see how well each model matches human variability at an item-wise level. We construct a pairwise dissimilarity matrix for the human ratings by taking the one minus the Pearson correlation between all ratings for each pair of videos (not just the median). In this way we can see how well each model captures the full variation of human

responses (now described in Results section “SocialGNN uniquely explains variance in human social interaction judgements”).

We find that both SocialGNN does a similarly good job of capturing the full range of human ratings (Figure 6) and interestingly explains unique variance in human judgements, even when accounting for variance explained by the SIMPLE model).

2. In the test accuracy calculation, it is not very proper to directly use the chance level as the baseline since machine learning algorithms are sometimes very powerful at getting results beyond the chance level even for some random data. Therefore, the shuffled baseline should be calculated. For example, labels of all the data in the dataset should be randomly shuffled and the same model with the same training process should apply the shuffled data. The process could be repeated thousands of times to estimate the shuffled baseline.

Thank you for this suggestion. Unfortunately, running full permutation testing versus chance, requires retraining each model thousands of time and is extremely computationally expensive (we estimated it would take ~60 days of run time). We ran 10 permutations for SocialGNN and found that like you suggest a few shuffles are above the chance level (accuracies across 10 shuffles: [0.41, 0.4, 0.39, 0.34, 0.38, 0.31, 0.35, 0.3, 0.35, 0.32]). They are all well below the performance of SocialGNN. Since the main claims of our paper center on model comparison, it does not seem worth the computational resources to run full permutation testing versus chance.

Based on your suggestion, we have now included permutation testing for all statistical inferences in the paper, including model comparisons and correlational analyses described above. (Methods lines 626-638).

3. I understand the authors used bootstrapped train/test splits. What is the number of iterations during bootstrapped?

Thank you for highlighting this omission. We ran ten bootstraps for Experiment 1 and twenty for Experiment 2 (to account for the larger variability in the distribution of labels in the Gaze dataset). We have now included these details in Results (lines 194, 288) and methods (lines 451-454 and 496).

4. Compared with Figure 4, the predicted accuracy of the inverse planning model in Figure 5 had a great improvement, reaching nearly 90% in Figure 5. As Figure 5 uses 100 new videos as the test set, which is a more difficult task. What is the reason the inverse planning model performed better under such conditions?

Thank you for highlighting this. Our results reported in the initial paper were comparing the inverse planning model to the social interaction labels included in the PHASE dataset rather

than our human ratings. We have corrected this error and now the model achieves around 83% performance (see Updated Figure 5A).

We were still surprised by the inverse planning's overall higher performance on the more difficult generalization set than main set. In follow up discussions with the PHASE paper authors they let us know that heuristics were added to their inverse planning model to better predict categories in the generalization set. These heuristics (based on video length) were not true in the main set, but the authors did not test model performance on the main set in their original paper.

We have omitted these details from the paper since they are not directly related to our main conclusions but are willing to add them in if the reviewers feel it is important.

5. Referring to Human Gaze Communication Dataset, why not ask human participants to rate these stimulus videos as human participants did for PHASE dataset?

Thank you for pointing out this omission. The Human Gaze Communication Dataset was released with human annotations (unlike the PHASE dataset which only included the label used by the physics simulator to generate the videos). Thus, we did not feel the need to collect additional ratings. We have now clarified this in the Methods section (lines 494-496):

Labels were collected by two human annotators in the original paper. As described in their methods, in videos where the two annotators disagreed a third "specialist" in the field assigned the label.

6. Confusion matrices of SocialGNN, VisualCNN and Inverse planning model did not report for PHASE dataset.

We have now included these in Supplemental Figures S9 and S10.

7. It is a bit surprising that the authors no longer compared SocialGNN with the inverse planning model in Human Gaze Communication dataset. Is there any particular reason for that?

In its current implementation, the inverse planning model is not image-computable. The model requires inversion of a generative model of the world (including a full physics simulator) which is not available (or tractable) in real-world videos. We explain this in the introduction and now clarify in this results section lines 308-309:

(Since the current Inverse planning model is not an image-computable model, it cannot be included in these experiments.)

8. The authors did not provide a run-time calculation for Human Gaze Communication dataset. It would be interesting to see this analysis.

We include this information below. Since only neural network models are evaluated on the gaze dataset, we did not include it in the main text, as differences between neural models in terms of computational resources consumed is not central to the claims of the paper. We are happy to add it in though, if the reviewer feels strongly.

	Memory		Run-time	
	2-way	5-way	2-way	5-way
SocialGNN	1.29MB-1.64GB*	1.29MB-1.66GB*	24-41s	24-40s
VisualRNN	1.29MB	1.29MB	12s	11-12s
VisualRNN-Rel	1.29MB	1.29MB	11s	11s
VGG19	5.48GB	5.48GB	38-40s	38-40s

* Amount of memory utilized varies across bootstraps due to differences in number of people/objects in videos

9. For figures with multiple subplots such as Figures 3, 5, and 6, I recommend labelling each subplot with Letters A B C to improve the readability.

These are now included in the top of subplot panels.

10. 46 lines are missing a period "recognizing social interactions Social interactions".

We have fixed this sentence.

Reviewer #2 (Remarks to the Author):

Summary and overall evaluation:

This paper presents a new computational model, SocialGNN, which captures how humans rely on relational visual information to recognize social interactions. I am not an expert on graph neural networks, so I cannot comment on the low-level implementation choices. But, if everything was executed correctly (and from what I can tell, it was), this paper is a breakthrough in both cognitive science and artificial intelligence.

In cognitive science, this is (to my knowledge) the first computational model of social computations in high-level vision. This is an incredibly important gap that needed to be filled. To date, we have models of high-level social cognition (e.g., meta-representational belief reasoning), and of low-level social perception (e.g., face detection and pose

estimation), but nothing in between. As consequence we understand little about the intermediate computations that interface low-level representations of agents with the machinery of high-level cognition. This paper fills this gap, and in doing so it opens the door to studying the full social reasoning pipeline computationally. This work has the promise of revolutionizing social neuroscience, in a similar way to how deep neural network models of object perception have revolutionized how we study the ventral stream.

This paper is also a great contribution to artificial intelligence. Despite the everyday breakthroughs in AI, these advances are usually limited to categorization, planning, and language, with surprisingly fewer advances in human-like social intelligence.

In short, I am very enthusiastic about this paper, and think it is a rare and valuable contribution.

Main areas requiring revisions:

I was surprised that the very limited space in the introduction was used to argue that inverse planning was wrong. I have no doubt that it is in many ways, but I did not see how this paper speaks to that issue. I do not think that the paper currently speaks to that and I wonder whether the conflict being set up in the introduction is an artificial one (and, as I see it, SocialGNN is a clear stand-alone contribution).

I believe this stems from a few related points:

1. Inverse planning is explicitly a framework for modeling high-level meta-representational Theory of Mind. The introduction frames this paper around the idea that inverse planning cannot be a model of visual perception, but I do not think that this a position that anyone holds.

The paper justifies this position by saying that people have used inverse planning to model how we infer social relations, but social relations are detected in high-level vision. The issue here is that inferences about social relations happen in both high-level vision and in high-level cognition. So, we need models of the inferences that happen at both levels, and I do not follow why the paper has the implicit assumption of that the inferences must either happen exclusively in high-level vision or exclusively in high-level cognition (and therefore, one of the models must be wrong).

I agree that the early modeling how ToM helps infer social relations used experimental paradigms that were too simple and that can be solved by vision alone (e.g., Ullman et al 2010). Therefore that work moved away from the simple helping/hindering paradigms to

include false-belief reasoning. If you consider, for instance, Hamlin et al., 2013, those tasks are clearly not solved by vision (see in particular the complex false belief conditions).

This makes me wonder whether the presented conflict between IP and SocialGNN is artificial. It might be more accurate to say that we have models of meta-representational Theory of Mind, but that there are a lot of social computations happening in high-level vision that we do not yet understand and that is the goal of this paper. The comparison against inverse planning is still very useful, but it is not obvious to me that the framing on “only one can be right” makes sense.

If there are papers that propose inverse planning as a model of visual perception, or argue that all inferences about social relations are driven by inverse planning, it would be helpful if this were cited in the introduction. In that case, it would be helpful to explain this as a position held by some people (since I think this is a fringe position), rather than an intrinsic component of inverse planning.

Thank you for your encouragement and these helpful suggestions. We agree that as previously written the paper introduction overstated the conflict between different modeling approaches. We agree with your later points that inverse planning models are models of theory of mind, the proposed GNN is a model of perception, and not vice versa. However, they are both models of social scene understanding, and the question at stake is whether humans are using more mentalistic or perceptual processes to recognize interactions.

We also agree that many (including us) believe humans engage in both perceptual and mentalistic processing of social scenes, and we need accurate models to reflect both types of processes. The major contribution of this paper is a high performing perceptual model of relational social scenes.

We have updated the introduction now to reflect these nuances. This section of the introduction now reads (lines 47-51 and 59-63):

While humans can clearly use high-level mental state inference to recognize and understand many aspects of social interactions (Kiley Hamlin et al., 2013; Woo et al., 2022), especially when visual cues are non-diagnostic, growing evidence suggests that social interactions are also rapidly recognized by the human visual system. ...

Despite this evidence, the field still lacks good bottom-up, visual models of social interaction recognition. Even deep learning models that achieve human-level performance in so many other visual tasks do a poor job modeling human social interaction judgments (Isik et al., 2020; Shu et al., 2021). It thus remains an open question to what extent humans rely on visual processes versus higher-level cognitive reasoning for social interaction recognition.

2. My understanding is that Inverse Planning is a proposal at Marr's computational level of analysis and is currently agnostic about the underlying inference algorithm. While some implementations use sampling, this is a computational placeholder for whatever inference mechanism might be implemented in the brain. For instance, one of the central papers proposing inverse planning (Baker et al., 2017) explicitly argues that compiling these inferences into a neural network architecture might be better for capturing the algorithmic level (like, e.g., Yildirim, et al., 2020).

As analogy, consider the proposal of perception as inverse graphics. Initial support for this idea was found through a first generation of models that aimed to test the value of the theory at a computational level of analysis, and then underwent a second generation implemented in a neurally-plausible way. The second generation of models did not imply that inverse graphics was wrong, and it would have been inaccurate to say that inverse graphics was intrinsically committed to top-down sampling just because this was a computational approximation of testing if the outputs of the two models matched.

In the context of this paper, some statements, like lines 36-39, are inaccurate. I don't think that the heart of the framework is to compare trajectories to internal simulations. Rather, it's the idea that the inference mechanism is approximating the inversion of a planner.

This makes the claims about run-time and memory load of inverse planning feel misleading. That being said, the memory load comparisons are still super important for discussing the value of SocialGNN in the context of AI. The paper just needs to be clearer about what's at stake in each claim.

Thank you for these comments we have modified the introduction to more accurately describe the motivation behind inverse planning systems, which we now describe as inverting a generative model of agents' interactions. We replaced the language about trajectory prediction with more general language about simulation and hypothesis generation. This section now reads (lines 36-41):

These models recognize social relationships by inverting a generative model of agent's interactions and comparing an observed social scene to internally generated hypotheses based on not only visual information, but also inferred physical information about the scene and hypothesized goals of the agents. So far, these models provide the best match to human judgments, suggesting humans rely on similar inferential processes to recognize interactions.

We agree with your points about memory and run time, and thus de-emphasized them in the framing in the introduction.

In the discussion, we now discuss the memory and run time results in terms of potential advantages for AI systems, not in the context of implications for human cognition.

3. With the two points above in mind, for the paper to deliver on the promise of the introduction, it would need to show that inverse planning is wrong, which requires two pieces of evidence that aren't in the paper yet.

First, you would need to show that SocialGNN is not a neural implementation of inverse planning.

For instance, you could show that SIMPLE and SocialGNN are uncorrelated, and that SocialGNN is more human-like on events where the models show the sharpest disagreement. I think this will be true based on the accuracy results shown in Figure 4, but it's still possible that this accuracy difference arises from relative confidence differences, and not qualitatively different computations. This argument could also be bolstered by analyzing the pattern of errors more clearly (more about this on point X below).

Although this isn't strictly necessary for publication, this would be a valuable addition. There is the caveat that I don't think anyone believes that inverse planning is a model of visual perception (more on this below).

Thank you for this suggestion. SocialGNN does not simply appear to be a neural implementation of SIMPLE. To test this at an item-by-item level, we performed representational similarity analysis (RSA) to compare pairwise responses on each video between models and human data. SocialGNN and SIMPLE are moderately correlated ($r=0.24$ and 0.4 on standard and generalization sets, respectively), and both are quite correlated with human data (Figure 6, left).

Interestingly, however, both models capture a significant amount of unique variance in human responses (semi-partial correlations, Figure 6, right) suggesting they are explaining different complimentary aspects of behavior.

We think this is very exciting because it suggests humans may switch between more perceptual and cognitive processes when judging social interactions. In addition, it opens the door to many future exciting modeling explorations to understand about how humans may combine these processes (and how this can be modeled using joint perceptual-inferential models).

These results are now presented in Figure 6 and lines 241-278, and their implications are discussed in lines 335-338:

Interestingly, SocialGNN is not simply a neural instantiation of the inverse planning model, and both models explain unique variance in human judgements, suggesting humans are using a combination of perceptual and mentalistic strategies to judge these videos.

Second, you would need to show that SocialGNN is better than inverse planning in the domains that inverse planning is proposed to work. For this, you would need to have false-belief style social inference tasks. My read is that the authors do not actually believe that this is the case. I found in particular the discussion to have much more nuance that was missing in the rest of the paper.

You are correct that we do not believe this is the case. SocialGNN is a model of social perception not social inference. What is at stake is the extent to which visual social interaction judgements can be made based on perceptual information alone (which has been debated), and whether we can model these abilities. We have aimed to clarify this in the introduction (lines 47-51 and 59-63, excerpted above) and discussion (lines 330-338, 370-373):

We developed a novel computational model, SocialGNN, that reproduces human judgments of social interactions in both animated and natural videos using only visuospatial information and bottom-up computation. This model performs as well as a generative inverse planning model and does so at a fraction of the computational cost without any explicit mental inference of agents' goals, suggesting that computations within the visual system may be sufficient for humans to recognize social interactions. Interestingly, SocialGNN is not simply a neural instantiation of the inverse planning model, and both models explain unique variance in human judgements, suggesting humans are using a combination of perceptual and mentalistic strategies to judge these videos.

These initial bottom-up, visual judgements may then be refined and supplemented by higher-level cognitive processing to give rise to the full range of humans' rich social scene understanding.

A few more areas where clarity would be helpful:

4. In the introduction the paper claims that inverse planning uses explicit representations of other agents' minds and that this conflicts with evidence that vision recognizes social interactions (lines 43-46). I did not follow this argument.

Is it not possible for social interaction to trigger explicit representations of minds within vision? It seems to me that a social representation of helping or hindering is, by definition, an explicit representation of a goal.

Thank you for pointing out this confusion. We have updated the introduction to clarify what we mean by "processed visually". Critically we do not mean visual information that is the processed by a downstream cognitive system or using cognitive computations. We now clarify this in the Introduction (lines 63-69).

Note here that by "visual" we do not mean that humans exploit very simple pixel-level information to recognize interactions. Nor, on the other hand, do we mean visual information that is processed by higher-level cognitive systems. Instead, the above evidence suggests that like other high-level features, including causality and animacy (Scholl & Tremoulet, 2000), social interactions are also processed within the visual system using spatiotemporal cues in a manner that is distinct from cognitive processing (Firestone & Scholl, 2016).

5. The discussion also states that SocialGNN does not have explicit goal representations, and that this implies that visual information alone is enough for humans to recognize social interactions (lines 296-300). I also did not follow this argument, for two reasons.

First, how do we know that SocialGNN is not explicitly representing goals? To show that, don't you need to show that there is no information in SocialGNN that exclusively represents goals?

By this we mean that information about goals is not given to the model in training and that it does not have this as part of a generative model of the world. It could learn to represent goals based on the training it receives but we would consider this to be an implicit representation of goals and/or the physics of the world.

Second, even if SocialGNN does not have explicit representation of goals, why does that imply that visual information alone is enough to recognize social interactions? Don't we already know that visual information alone is enough? Wouldn't this be true independent of whether the inferences happen in high-level cognition or not?

Thank you for pointing this out. We have clarified based on your suggestions so this section now reads:

“...videos using only visuospatial information and bottom-up computation. This model performs as well as a generative inverse planning model and does so at a fraction of the computational cost without any explicit mental representation of agents' goals or the physical world, suggesting that computations within the visual system may be sufficient for humans to recognize social interactions.”

We hope that this in combination with the added details in the introduction clarify this distinction.

6. “In the past, the hypothesis that humans use only bottom-up, visual information to recognize social interactions has been dismissed due to the poor performance of purely visual models.” (lines 308-309).

I was not aware of this. Are there any references that could be added?

We have added these references, and briefly discuss them in the introduction (lines 60-62):

Even deep learning models that achieve human-level performance in so many other visual tasks do a poor job modeling human social interaction judgments (Isik et al., 2020; Shu et al., 2021).

7. I found it difficult to get a sense of what the models were doing based only on the abstract metrics. Is it possible to include a sub-set of stimuli/trials with the corresponding inferences from each model. It would be helpful to get a richer sense of what the models looked like in a more trial-by-trial basis.

Thanks for this suggestion. We have now included four example stimuli (Videos S1-S4) one for each case when the two models agree or disagree with each other and agree or disagree with human ratings. We hope these examples help provide some intuition.

We also ran some analyses to try to understand the types of videos that the models agree versus disagree on and match human judgements (see below plots of video counts where both, socialGNN only, SIMPLE only, or neither model matches the median human judgement). We found some preliminary evidence that SIMPLE tends to do a better job of classifying adversarial videos, and SocialGNN a better job on friendly and neutral videos, but these trends were prominent across both the Generalization set (left) and Standard or main set (right). We ran similar analyses separating the PHASE videos by goal type or human agreement, but neither of these revealed clear differences between the two models. For the sake of clarity, we have decided to leave these analyses/summaries out of the paper but are willing to put them back in (or consider other ways to show trial-by-trial behavior) if you have further suggestions.

8. The main text alludes to the fact that SocialGNN makes more human-like errors and directs readers to supplemental information. The text here is pretty vague and there is no proper analyses of the errors. To keep that claim, you might want to extend this into a proper and detailed statistical analysis.

We agree that the results previously reported in Figure S5 were inconclusive, since we did not have enough misclassified videos from either model to make a statistical claim. We have removed this figure and sentence from the main text. We believe the RSA analysis does a more compelling job of showing how both models explain not only consensus human ratings but also variability in human judgements (Results and Figure 6).

Minor:

- Missing period in line 46.

This has been added.

- I could not understand the sentence in lines 56-58.

This sentence has been reworded.

- Inverse Planning is a general framework, but the paper tests a very specific instantiation of it. That's fine, but a lot of the metrics reported report idiosyncrasies about a very specific implementation. It would be more accurate to use the term SIMPLE in the results and figures (the run-time and memory values reported in Fig. 5 are about SIMPLE, and not a general signature of inverse planning).

Thank you for pointing this out. We have emphasized this when the model is first introduced (lines 189-192) and in the figures. We have opted to leave the term "inverse planning" in the main text for readability (we worry the term SIMPLE is confusing and hard for the reader to keep track of), but are open to changing it if the reviewer feels very strongly.

we also compared SocialGNN and the baselines to the performance of an instantiation of an Inverse Planning model, SIMPLE, which currently achieves state-of-the-art performance on this task (Netanyahu et al., 2021).

- Discussion on lines 309-312 might be inaccurate. It pits inferences in vision as conflicting with an innate moral code. Why can't it be both? I think the community arguing for an innate moral code would be perfectly happy with an account where vision detects first-order relations (friendly, adversarial, etc), which are then supplemented by high-level cognition (e.g., seemingly friendly but with bad intentions; seemingly adversarial but they're looking out for their friend from making a bad choice; etc).

You are right. We have made a minor edit to clarify ("not using visual information" → "not based on visual processes") and added a sentence at the end of the paragraph to explain how these two systems might interact.

These initial bottom-up, visual judgements may then be refined and supplemented by higher-level cognitive processing to give rise to the full range of humans' rich social scene understanding.

- One other small point is that the paper often talks about 'inferences based on visual information' when I think it means to say 'inferences happening within the visual system'. I found this to be about confusing.

Thank you. We have tried to clarify this by replacing the terms “based on visual information” throughout the paper. We hope these re-wordings plus the addendum to the introduction mentioned above helps to clarify this distinction.

- While the paper is correct that inverse planning in its most general formulation is intractable, it actually does become tractable when situational constraints are added. See Introduction claims that inverse planning is intractable, but it isn't! See Blokpoel, Kwisthout, van der Weide, Wareham, & van Rooij (Journal of Mathematical Psychology).

Thank you for this pointer and suggestion. We have qualified this sentence in the introduction and added a reference to this work (lines 43-44).

Summary of suggested changes and final remarks:

This is a fantastic paper and all concerns listed above can be addressed. There are multiple types of revisions that would satisfactorily address these comments, but here are two possible routes:

1. The case against inverse planning.

The paper could be revised to make a tighter case against inverse planning and show stronger evidence against it. To achieve this, you would need to identify the special position of inverse planning as a model of high-level vision and show that SocialGNN is not a compiled version of inverse planning, but a fundamentally different type of computational process. Based on the presented results, I am very optimistic this would come out.

2. Filling the gap in modeling social intelligence.

The paper could be revised to highlight the gap that SocialGNN fills. We have models of mid-level vision, and of meta-representational social cognition, but there are these rich inferences performed in vision and we have no models for that, and here is where SocialGNN. Critically, this version should still include the inverse planning comparisons, but consider using it as a proxy to show that these are not high-level ToM inferences (rather than saying that only one model can be right). In this case, I think it would still be valuable and important to show that SocialGNN is not an implementation of inverse planning, although it is not critical for publication.

Again, this is impressive work!

Thank you for your positive and constructive feedback. We believe the manuscript is much more compelling, clear, and nuanced based on your suggestions. We hope you agree!

Reviewer #3 (Remarks to the Author):

Review

Summary of paper: The paper proposes an algorithmic-level model based on graph neural networks (GNNs) to explain human social inferences. The GNN model (referred to as SocialGNN) takes as input spatial and physical information about a scene (e.g. the positions of objects, or gaze directions), performs a single step of relational inference at each timestep, aggregates the relational inferences across time via an RNN, and then decodes the result into a social judgment. This model is compared to two alternative models (a non-relational RNN, and a Bayesian Inverse Planning model) across two tasks (the PHASE dataset, which is a 2D animated dataset of Heider-and-Simmel-like scenes; and the Human Gaze Communication dataset, which involves naturalistic videos with annotated gaze/person/object information). SocialGNN achieves higher test set accuracy than either the RNN baseline or Inverse Planning model, and also lower computational cost than the Inverse Planning model. The paper concludes that SocialGNN is a better model of human social inferences, and as such, lends support to the hypothesis that social inferences may be performed bottom-up via perception and do not always require planning/mental simulation.

Summary of review: This paper studies an important topic (explaining human social inferences) and investigates a promising class of models from machine learning. I think this is really important research to do, and it's fantastic to see work which tackles this! In particular, while existing models (like the Inverse Planning model) are great at explaining human judgments, they are not always algorithmically very plausible, are very much limited in their generalizability due to their computational burden, and can't be easily applied when symbolic representations aren't available. As such, identifying algorithmic-level models which can more efficiently and effectively implement computational-level goals is a pressing concern in computational modeling of social cognition. However, while I laud the paper's goals, I believe it falls somewhat short on a few dimensions regarding the model's architecture, analysis, and baselines (see detailed comments below). I believe this could be a great paper, but also that it needs a bit more work.

Detailed comments, roughly in order of importance:

The paper claims that SocialGNN "generalizes" to different settings, but to achieve this, three different variants of SocialGNN are required (small edge update model without context, larger edge update model with context, and node update model). It feels like a stretch to say that the same model is generalizing across these different settings because these are really two different models under the most generous interpretation, maybe three under a less generous interpretation. (It would be much more compelling if exactly the same

architecture were used, especially in terms of using the same edge/node updates across the PHASE and Gaze datasets, and in terms of using the same context across the two splits of the PHASE dataset).

Thank you for this suggestion. In the prior version of the paper the same architecture (SocialGNN-V) was implemented in all three experiments. We have now added a larger version of this model to the standard PHASE dataset experiments and find that the same architecture (same size node model) performs well in all datasets. See updated Figures S4 and 7.

We continue to report the SocialGNN-V (node) results for PHASE in the supplement and SocialGNN-Edge (which we have similarly updated with larger size and context, see Figure 4) in the main text since we believe the edge model is more straightforward to explain. The two models perform very similarly, and we can make the same claims with either so are happy to switch the order, but we worry it would affect the readability of the paper.

We did not implement SocialGNN-E in the gaze dataset because the human annotations included in the dataset were collected for each person in each video. Since the number of people can vary video to video it is not straightforward to convert these to a fixed number of edge ratings.

We do note though that while SocialGNN-V used for the gaze experiments is the same architecture as PHASE, this model still does not have context. Context is likely to help in these natural settings, particularly for transfer learning, but it is not straightforward to decide what form this should take. We think this is an interesting area for future research that we now discuss in lines 351-359:

In addition, while the same model architecture generalized across both animated and natural videos, we did not test a single trained model's ability to generalize, due to the different interaction types in the two video datasets. However, the model receives relatively little training data in each experiment. We believe with the right datasets, transfer learning across very visually different scenarios would work well, particularly if the right type of context information is added for each dataset. Determining optimal context variables for different settings is an interesting area for future research that could further improve SocialGNN's ability to generalize.

The analyses in the paper focus only on comparing average accuracy across models. This is a fairly weak measure of how well the models explain human cognition, as it does not capture any nuances of the behavior. For example, how well does the model explain individual scenes? Is it more unsure on scenes that have higher disagreement amongst humans? etc. (In analyzing the PHASE dataset, the variance across human judgments is discarded, which is a shame---this is rich information that would be really revealing when

compared to the model. For example, for each scene you could compute the KL divergence between the distribution of human judgments and the distribution/logits produced by each model. The paper's claims would be much stronger if you could show that the model is uncertain on scenes in which people are uncertain, and more certain when people agree. I realize there is some analysis along these lines in Fig S5, but this just looks at whether the model is correct/incorrect as a function of human agreement with the mode, which is not the same thing as comparing the distributions of the model and of people. It's also not obvious to me that the difference between the models in this figure is statistically meaningful.)

Thank you for this suggestion. SocialGNN does not simply appear to be a neural implementation of SIMPLE. To test this at an item-by-item level, we performed representational similarity analysis (RSA) to compare pairwise responses on each video between models and human data. SocialGNN and SIMPLE are moderately correlated ($r=0.24$ and 0.4 on standard and generalization sets, respectively), and both are quite correlated with human data (Figure 6, left).

Interestingly, however, both models capture a significant amount of unique variance in human responses (semi-partial correlations, Figure 6, right) suggesting they are explaining different complimentary aspects of behavior.

We think this is very exciting because it suggests humans may switch between more perceptual and cognitive processes when judging social interactions. In addition, it opens the door to many future exciting modeling explorations to understand about how humans may combine these processes (and how this can be modeled using joint perceptual-inferential models).

These results are now presented in Figure 6 and lines 241-278, and their implications are discussed in lines 335-338:

Interestingly, SocialGNN is not simply a neural instantiation of the inverse planning model, and both models explain unique variance in human judgements, suggesting humans are using a combination of perceptual and mentalistic strategies to judge these videos.

I believe the paper uses the term "visual" incorrectly. For the PHASE dataset, SocialGNN does not take visual information as input: it takes spatial information. Visual information would be, for example, pixel values. An object's location, velocity, size, etc. are not visual properties but are physical or spatiotemporal properties. This may seem like a nitpick, but I think it's an important distinction because inferring spatiotemporal properties from visual input is a nontrivial task and is still, in some cases, an unsolved problem in computer vision. Stating that SocialGNN can infer social information directly from visual input is therefore attributing it more ability than it actually has. (For the Gaze dataset, it's somewhat more accurate because the CNN takes visual information as input, though it is still given object locations and sizes in the form of bounding boxes, as well as gaze directions, which are spatial/physical information). Here are some locations I noticed where the term "visual" is used incorrectly: in the title, the names "VisualRNN" and "VisualRNN-Rel", lines 64, 71, 119, 129, 195, 196, 221, 230, 296, 299, 319, 348, 359, 373, 444, 490, and 502.

You are correct that we did not put pixel-level information into the PHASE dataset experiments. However, we believe, particularly in the datasets we use, this is largely a solved computer vision problem. The PHASE dataset is a very simplified visual stimulus so extracting properties like the shape, size, speed, and orientation of the entities would not be difficult (in fact many of these properties could be extracted based on object color alone). For the Gaze dataset, automatic computer vision systems currently do a good job of detecting bounding boxes and gaze direction (we have used the Amazon Rekognition platform for similar purposes, and there are also several classifiers for face and eye detection in OpenCV). While we believe it would be fairly straightforward to extract these properties in the current paper, we don't think it meaningfully adds to the intellectual contribution of the paper.

More generally, we think it is important to clarify what we mean by "processed visually". Throughout the paper we use this term not to refer to simple pixelwise information, but instead to refer to the distinction between visual perception and cognition. We now clarify this in the introduction, lines 63-69.

Note here that by "visual" we do not mean that humans exploit very simple pixel-level information to recognize interactions. Nor, on the other hand, do we mean visual information that is processed by higher-level cognitive systems. Instead, the above

evidence suggests that like other high-level features, including causality and animacy (cite Scholl & Tremoulet, 2000), social interactions are also processed within the visual system using spatiotemporal cues in a manner that is distinct from cognitive processing (Firestone & Scholl 2016).

We hope this clarifies our use of the word “visual” in the paper, particularly in the context of “visual representations” or “visual models” (both in contrast to “cognitive” representations or models). We do agree with your point though that the terms “visual information” and “purely visual” in the text may be misleading, so have changed these terms to “visuospatial information” or “bottom-up”.

The detailed results with the RNN baseline---in which it collapses into making constant predictions---suggest to me that there is a bug. My suspicion would be that the model size is not large enough to capture the information in the task. It's also not really the most compelling baseline, to be honest, because concatenating all the object information and feeding it through an LSTM is an atypical choice and one that I wouldn't guess would work well for *any* task. (A better baseline would be a CNN or Vision Transformer architecture). Regardless, it should at least be able to memorize the training data; if it cannot do that, then it is not expressive enough.

For the PHASE dataset, the inverse planning model was introduced as the state-of-the-art benchmark. A very similar model to our VisualRNN (2-Level LSTM) was the next best model of relationship prediction so we believe the comparisons to be fair on this benchmark, and we now mention this when we introduce the model in the Results section (lines 184-185):

VisualRNN is an implementation of a Cue-Based LSTM, a standard perceptual baseline used in similar tasks (Netanyahu et al., 2021; Shu et al., 2020).

We take your point though that the VisualRNN models are not well suited to the natural videos in the gaze dataset. We find that to be a compelling part of our results. A small and well understood tweak (the addition of graph input and processing) to our model leads to a large improvement in accuracy.

To address your concerns, we first performed additional fine-tuning on these models to optimize them for natural images. This led to improvements in their accuracy and the models are no longer outputting single class labels (See updated Figure 7 and Supplemental Figures S7 and S8). However, both models still perform significantly worse than SocialGNN.

Based on your suggestion, we next tested a more standard CNN model typically used to recognize information in natural images, VGG-19. We include a justification for this model selection in Methods (lines 584-599)

We also compared our SocialGNN model with a standard CNN on the natural videos (Gaze dataset). We took a pretrained VGG19 (pretrained on ImageNet, (Simonyan & Zisserman, 2014)). Although this model was trained to recognize objects, it has been shown that its learned representations generalize to different high-level visual tasks with fine-tuning (Geirhos 2021, Schrimpf 2018). We selected for this model that operates on images (in our case frames of each video) rather than videos, since CNNs that operate on dynamic input have many more trainable parameters and have been shown to have worse cross-task generalization (Kataoka 2020). We input pixel information from each entire frame of each video to the model. The output from the penultimate fully connected layer is then taken and reduced to 1500 dimensions using PCA. We averaged these features across frames for each video clip and passed it through a linear layer that we train to either do the 2-way or the 5-way classification on the Gaze dataset using the same cross-validation procedure as SocialGNN and the RNN models. While we do not have the scale of dataset to relearn all the model weights via backpropagation, our procedure is equivalent to fine-tuning the model on our tasks. See Supplementary Section: Experimental Settings for trainable parameter settings such as learning rate, regularization parameter and class weights.

Despite its success in matching human behavior across a range of other domains, it does a poor job of matching human social interaction judgements. While there are more advanced CNNs available, they in general don't do a much better job of explaining human behavior across high-level visual tasks (e.g., Geirhos 2021, Schrimpf 2018). While it is possible that with enough data end-to-end training of a vision transformer or CNN would do a better job at this task, unfortunately we don't have access to a labeled dataset at that scale. We also think that the relative success of SocialGNN highlights the efficiency of using a relational inductive bias in these low-training data settings.

From a machine learning perspective, the SocialGNN architecture is rather atypical and I would be surprised if it works for more complex tasks. Perhaps that is fine for a cognitive model, but it might limit the generality of the model if it won't work in more complex settings. Could you please elaborate on how you chose the particular architecture, what alternatives you tried, and how they performed? In particular, why not use a more typical GNN architecture, e.g. a recurrent GCN or recurrent MPNN?

Thank you for these suggestions. First, we would like to clarify that we do see SocialGNN primarily as a cognitive model, but one that can be scaled up for more complex tasks (Battaglia et al., 2018) and that we hope will be useful to the AI community.

We based our model's architecture on a standard perceptual benchmark in this area of cognitive modeling (VisualRNN, which is often referred to as a cue-based LSTM) and added graph structure and representations on this LSTM backbone. This worked quite well on our initial benchmark task, where other more standard graph nets have failed (Netanyahu et al., 2021, see below). It also allowed for the most controlled model manipulation between our benchmark model (matched except for the addition of graph input and processing) to most

clearly answer our scientific question, so we did not explore other graph architectures.

In the Gaze dataset, how are the labels determined? The paper remarks upon "the importance of using human ratings rather than the default labels" (line 108) for the PHASE dataset, but then seems to rely on the default labels for the Gaze dataset. Were these labels similarly determined from multiple human judgments? As it's currently presented, the results on this dataset don't appear particularly relevant to validating models of human cognition because I am not sure how much they're actually capturing about human judgments vs. just whatever the annotator said. At a minimum, it would be helpful to discuss how the labels were collected (e.g. how many judgments per label, what the population was, etc.). As discussed above, it would be even better if it were possible to perform some more fine-grained analyses (e.g. does the model's distribution match the distribution of human ratings on individual stimuli).

Thank you for pointing out this omission. The gaze dataset labels came from human annotators from the original paper (unlike the main PHASE set which was the output of a physics simulator). In videos where the annotators disagreed, labels were assigned by a third "specialist" in the area. We now describe this in the methods (lines 493-496).

The gaze communication labels provided include "non-communicative", "mutual gaze", "gaze aversion", "gaze following", and "joint attention". Labels were collected by two human annotators in the original paper. As described in their methods, in videos where the two annotators disagreed a third "specialist" in the field assigned the label.

One limitation of this approach, however, is that the labels don't reflect the inter-subject variability of human judgements. We see the natural video experiments in our work as an initial proof of concept and believe that future work should investigate the extent to which SocialGNN and other models capture not only accuracy of human judgements, but also variability in natural videos.

Lines 199-200: "SocialGNN even outperforms the Inverse planning model on the standard PHASE dataset." I'm not sure this is a meaningful comparison. SocialGNN is trained in a supervised way to predict human judgments and tested on similar (in-distribution) scenes. Given an expressive enough NN model class, any NN trained on this data will be able to do well on in-distribution scenes. In contrast, the Inverse Planning model is not given the same training data and is able to explain human judgments instead given a few assumptions about planning/goals/etc. That doesn't invalidate the utility of a model like SocialGNN--- especially when compared to other NN architectures trained on the same data, it suggests which architectural features contribute to sufficient expressivity---but it does feel a little bit like comparing apples to oranges to compare it to the Inverse Planning model in this way. (I think it's telling that when generalizing out of distribution, the Inverse Planning model does much better than SocialGNN).

Thank you for raising this point. While not trained on human data, the inverse planning model is given information about physical and social characteristics of world through its physics simulator and the model has access to dataset heuristics in its priors. On the other hand, SocialGNN does have direct access to human ratings, though on separate videos. In addition, SocialGNN is trained on relatively little data. However, we acknowledge this as a shortcoming of neural models that we now mention in the discussion (lines 353-357):

Unlike the inverse planning model, SocialGNN (like the other neural models tested here) was trained on human data. However, the model receives relatively little training data in each experiment. We believe with the right datasets, transfer learning across very visually different scenarios would work well, particularly if the right type of context information is added for each dataset.

Finally, while the inverse planning model does have higher accuracy on out of distribution data, the RSA analysis revealed that both models perform similarly at matching item-wise human judgements (see Figure 6).

How many model runs/seeds were used to compute the results?

Thank you for highlighting this omission. We ran ten bootstraps for Experiment 1 and twenty for Experiment 2 (to account for the larger variability in the distribution of labels in the Gaze dataset). We have now included these details in lines 451-454 and 496.

Lines 335-338: "SocialGNN is inspired by a growing body of recent work in graph neural network modeling for social behavior and multi-agent systems (Shu et al., 2021; Sun et al., 2022; Tacchetti et al., 2018). Unlike SocialGNN, however, these models all seek to predict agent trajectories, either as their final output or as an intermediate step towards a final social prediction." If there are existing NN-based cognitive models of social predictions, I would think they should be included for comparison, even if they predict agent trajectories as an intermediate step: isn't one of the goals of the paper to demonstrate that such predictions aren't necessary? To validate this claim, comparisons should be included to similar NN models which do include such assumptions.

In the initial PHASE paper, the authors use the ARG model (Wu et al., CVPR 2019), a graph net which uses trajectory prediction as an intermediate step for social activity recognition and find that it performs quite poorly (worse than the cue-based LSTM). Further, as mentioned above the current architecture allows us to have a fairly tight model comparison to the previous perceptual baseline strengthening our claims about the importance of graph processing in particular. In more standard graph-nets it would likely be difficult to disentangle the role of many differing factors between those models and cue-based LSTM (aka Visual-RNN).

More generally though, your comment highlights that we should clarify that we are not making strong claims about the role of trajectory prediction in these processes, but instead arguing against hypothesis generation and inverse inference on explicit mental models of other agents. We think this confusion stemmed from our original introduction which conflated the two processes. We have now clarified our argument in the introduction (lines 36-41):

These models recognize social relationships by inverting a generative model of agent's interactions and comparing an observed social scene to internally generated hypotheses based on not only visual information, but also inferred physical information about the scene and hypothesized goals of the agents. So far, these models provide the best match to human judgments, suggesting humans rely on similar inferential processes to recognize interactions.

Trajectory prediction may be helpful in addition to or as part of a perceptual model, but we believe answering that question is outside of the scope of our claims and investigation.

Lines 348-349: "giving standard visual models relational information (as in the case of VisualRNN-Rel) is not enough to reproduce human behavior". I think it's a stretch to call VisualRNN a "standard visual model". A standard visual model would be a CNN. The architecture of VisualRNN is actually quite unusual, as I discussed above.

We have changed this sentence to read:

simply giving models relational information (as in the case of VisualRNN-Rel) is not enough to reproduce human behavior

Lines 356-358: "it uses the same set of functions across all nodes and all edges, making the model capable of combinatorial generalization in a human-like manner". This statement is unsupported; I don't think the paper has demonstrated that this model achieves combinatorial generalization.

This claim is made in the Battaglia et al. 2018 paper cited at the end of the sentence, but since it is not based on the results of the paper we have softened the language and added the reference:

Further, it uses the same set of functions across all nodes and all edges, which may make the model capable of combinatorial generalization in a human-like manner (Battaglia et al., 2018).

Lines 411-412: "Participants who did not rate all three catch trials correctly were excluded

from further analysis". How many people were excluded in total?

108 participants were excluded in total, including those who did not complete the experiment. We have added this detail to the paper.

Lines 37-39: "These models infer social relationships by comparing the observed agent trajectories to internal simulations based on not only visual information, but also inferred physical information about the scene and hypothesized goals of the agents." To be fair, SocialGNN also provides inferred physical information about the scene. It also is given access to information about what different types of social interactions look like, in the form of its training data, which other models do not have access to (as discussed above).

Thank you for pointing this out. In addition to the caveat discussed above, we have removed claims that SocialGNN does not represent physical information, and added the above caveat about training on human labels. On the other hand, we would argue that the full physical simulations available to the inverse planning model are much more extensive than the simple spatial features provided to SocialGNN.

Lines 56-58: this sentence is ungrammatical.

Thank you. We have fixed this sentence.

Reviewers' Comments:

Reviewer #1:

Remarks to the Author:

The authors have addressed my major concerns satisfactorily. There is only a minor point to consider.

Regarding the generalization of the model, it is essential to adopt more cautious language when discussing it, as there is currently no evidence of transfer learning between two different datasets.

The Discussion (Line 356-359) "

We believe with the right datasets, transfer learning across very visually different scenarios would work well, particularly if the right type of context information is added for each dataset"

This discussion may lead readers to question whether the authors utilized appropriate datasets in their paper.

Would it be better to state that there is currently insufficient evidence to support the claim that the model exhibits transferability across different databases? Instead, it highlights the potential for future research to explore more diverse datasets and context information to improve model generalization.

Reviewer #2:

Remarks to the Author:

I have read the revised manuscript and all my concerns were addressed. The new analyses showing that SocialGNN is not a neural implementation of inverse planning really strengthen the paper and further increase the impact of this work. I am very enthusiastic about this paper and I believe it is an important step towards building models of complex social perception.

I only have some minor optional suggestions that do not affect my recommendation:

1. I was left feeling uncertain about what exactly SocialGNN is ultimately a model of. A big part of the paper's motivation is that social relations are detected within perception, which sounds like SocialGNN is a model human social perception. On the other hand, the revision emphasizes optimizing features through engineering, which sounds like this is not meant to be a biological model, but an engineering advance, loosely inspired by humans. A third possibility is that this is meant to be proof-of-concept that social relations can be computed without top-down hypothesis search, require no commitment about biological plausibility or engineering potential.

This comment as minor because any of these angles is an important advance. But I would find it helpful if the paper were more explicit about how exactly the authors conceptualize the model. For instance, would you predict that SocialGNN should have predictive power over neural computations? It would be useful if the conceptualization had enough information for someone to know what type of data would falsify the model (if there is a strong analogy to social perception) or if the model is falsifiable (which it may not, if this is meant to be proof-of-concept that certain computations are possible withing GNNs).

2. The writing on lines 15-17 in the abstract seem to imply that visual processes are separate from complex mental inference. I was surprised because I would have thought that the authors would agree that perception performs all sorts of complex mental inferences. In some ways, the results from this paper further make this point, showing that perception is much more cognitive than previously thought.

3. Some references that may or may not be useful: You might want to consider citing the 'original' papers by Chris Baker first proposing the inverse planning account in a fleshed out. Baker et al., 2009 (formalizing goal inference through MDPs) is probably the most relevant.

Another paper that might be relevant (though perhaps no clear place to cite) is Lindsey Powell's work on adopted utility calculus, which presents an overarching theory of social relations through utility-focused inverse planning.

5. There are extra parentheses closing on lines 44 and 59.

References:

Baker, C. L., Saxe, R., & Tenenbaum, J. B. (2009). Action understanding as inverse planning. *Cognition*, 113(3), 329-349.

Powell, L. J. (2022). Adopted utility calculus: Origins of a concept of social affiliation. *Perspectives on Psychological Science*, 17(5), 1215-1233.

Reviewer #3:

Remarks to the Author:

I would like to thank the authors for their extensive revisions. I believe the paper is now greatly improved and am happy to recommend acceptance.

In my original review, I wrote: "I believe it falls somewhat short on a few dimensions regarding the model's architecture, analysis, and baselines". I believe these concerns have been suitably addressed, which I will elaborate on below:

- Regarding the generality of the architecture: thank you for adding the experiments with SocialGNN-V, I believe this makes a much more compelling case for the generality of the model.

- Regarding more finegrained analysis: thank you for adding the RSA analysis! I think this is really informative and makes the claims much stronger. As a way to make the particular claim about SocialGNN and SIMPLE being complimentary, it could be really interesting to try a hybrid model that uses a weighted combination of their predictions to see if that can explain human performance even better than either alone. (This would be, I think, somewhat more compelling than the semi-partial correlation analysis, though that already I think is helpful).

- Regarding the term "visual": thank you for updating the paper to be more precise in the terminology, I think it is much clearer now. Though just to remark on one comment: "we believe... this is largely a solved computer vision problem". As someone working in machine learning, it is definitely not! I agree simple solutions might work well in the PHASE dataset so perhaps it could be considered "solved" in that case, but I'd be very careful about claiming too much about what computer vision models can do in the general case without actually demonstrating it.

- Regarding the RNN: thank you for clarifying that it is a pre-existing model, and for performing the additional fine-tuning to fix its performance. From the ML perspective though, I still think it is a strange architecture and not one that would work well, so it still feels to me like a strawman baseline and that more meaningful baselines could be devised. But, I am glad to see the addition of the CNN baseline and am satisfied with that addition.

- Regarding the SocialGNN architecture: that's fair to have chosen the architecture for controlled manipulation. I would encourage you though in future work to look at more canonical and battle-tested GNN architectures, such as GCN (graph convolutional networks) or MPNN (message-passing neural network) as I mentioned. In particular, using a recurrent MPNN would address the difficulty you alluded to earlier in the rebuttal: "We did not implement SocialGNN-E in the gaze dataset because the human annotations included in the dataset were collected for each person in each video. Since the number of people can vary video to video it is not straightforward to convert these to a fixed number of edge ratings.". MPNNs have no requirement for a fixed number of edges.

- Thank you for the remaining clarifications and edits.

REVIEWERS' COMMENTS

Reviewer #1 (Remarks to the Author):

The authors have addressed my major concerns satisfactorily. There is only a minor point to consider.

Regarding the generalization of the model, it is essential to adopt more cautious language when discussing it, as there is currently no evidence of transfer learning between two different datasets.

The Discussion (Line 356-359)“

We believe with the right datasets, transfer learning across very visually different scenarios would work well, particularly if the right type of context information is added for each dataset“

This discussion may lead readers to question whether the authors utilized appropriate datasets in their paper.

Would it be better to state that there is currently insufficient evidence to support the claim that the model exhibits transferability across different databases? Instead, it highlights the potential for future research to explore more diverse datasets and context information to improve model generalization.

Thank you for your support and this suggestion. We agree that this is an important area of future research to highlight. We also take your point that this point is largely speculation, and now explicitly acknowledge this limitation as you suggest. At the same time, we want to make it clear that we believe this is a promising avenue for future work. To further address your concern, we have updated our current wording to stress the fact that this is our opinion, rather than an empirical claim, and highlight the need for future work.

In addition, while the same model architecture generalized across both animated and natural videos, we did not test a single trained model's ability to generalize, due to the different interaction types in the two video datasets. As a result, there is currently no direct evidence that transfer learning will work in SocialGNN. Further, unlike the Inverse Planning model, SocialGNN (like the other neural models tested here) was trained on human data. However, the model receives relatively little training data in each experiment. We believe with the right datasets, transfer learning across very visually different scenarios may work well,

particularly if the right type of context information is added for each dataset. Transfer learning and optimizing context variables for different settings are interesting areas for future research that could further improve SocialGNN's ability to generalize.

Reviewer #2 (Remarks to the Author):

I have read the revised manuscript and all my concerns were addressed. The new analyses showing that SocialGNN is not a neural implementation of inverse planning really strengthen the paper and further increase the impact of this work. I am very enthusiastic about this paper and I believe it is an important step towards building models of complex social perception.

I only have some minor optional suggestions that do not affect my recommendation:

1. I was left feeling uncertain about what exactly SocialGNN is ultimately a model of. A big part of the paper's motivation is that social relations are detected within perception, which sounds like SocialGNN is a model human social perception. On the other hand, the revision emphasizes optimizing features through engineering, which sounds like this is not meant to be a biological model, but an engineering advance, loosely inspired by humans. A third possibility is that this is meant to be proof-of-concept that social relations can be computed without top-down hypothesis search, require no commitment about biological plausibility or engineering potential.

This comment as minor because any of these angles is an important advance. But I would find it helpful if the paper were more explicit about how exactly the authors conceptualize the model. For instance, would you predict that SocialGNN should have predictive power over neural computations? It would be useful if the conceptualization had enough information for someone to know what type of data would falsify the model (if there is a strong analogy to social perception) or if the model is falsifiable (which it may not, if this is meant to be proof-of-concept that certain computations are possible withing GNNs).

Thank you for your support and for raising this concern. We think of SocialGNN as a combination of 1 (model of human social perception) and 3 (proof-of-concept that social interactions can be recognized without top-down hypothesis search), though we believe these findings could be used for 2 (as an engineering advance). To point 1, it is a representational (though not necessarily "biologically plausible") model of human social

perception that we think should match neural data (based on prior work from our group, specifically in the pSTS and that this would be a way to falsify the model. We think it may be adopted as an engineering advance but that is not the main goal or contribution of the paper. We have clarified this by adding the following in the Introduction (lines 74-75):

The result is a novel graph neural network (GNN) for social interaction prediction that we call SocialGNN, which serves as a representational level model of human social perception.

We also discuss this point and mention possible match to neural data briefly in the Discussion section (lines 431-433):

SocialGNN and the results presented here provide a new computational framework to answer these questions and understand the neural and behavioral representations of social interaction scene understanding.

2. The writing on lines 15-17 in the abstract seem to imply that visual processes are separate from complex mental inference. I was surprised because I would have thought that the authors would agree that perception performs all sorts of complex mental inferences. In some ways, the results from this paper further make this point, showing that perception is much more cognitive than previously thought.

You are right, we do believe that perception involves making all sorts of complex inferences. However, we do not think it involves complex *mental* inferences / making inferences about others' mental states. We have clarified this in the abstract:

However, growing behavioral and neuroscience evidence suggests that recognizing social interactions is a visual process, separate from complex mental state inference.

3. Some references that may or may not be useful: You might want to consider citing the 'original' papers by Chris Baker first proposing the inverse planning account in a fleshed out. Baker et al., 2009 (formalizing goal inference through MDPs) is probably the most relevant.

Another paper that might be relevant (though perhaps no clear place to cite) is Lindsey Powell's work on adopted utility calculus, which presents an overarching theory of social relations through utility-focused inverse planning.

Thank you for these suggestions. We now cite both papers in our Introduction section.

5. There are extra parentheses closing on lines 44 and 59.

We believe this may be due to a reference formatting error in an earlier version of that paper and it has been fixed.

References:

Baker, C. L., Saxe, R., & Tenenbaum, J. B. (2009). Action understanding as inverse planning. *Cognition*, 113(3), 329-349.

Powell, L. J. (2022). Adopted utility calculus: Origins of a concept of social affiliation. *Perspectives on Psychological Science*, 17(5), 1215-1233.

Reviewer #3 (Remarks to the Author):

I would like to thank the authors for their extensive revisions. I believe the paper is now greatly improved and am happy to recommend acceptance.

In my original review, I wrote: "I believe it falls somewhat short on a few dimensions regarding the model's architecture, analysis, and baselines". I believe this concerns have been suitably addressed, which I will elaborate on below:

Thank you for your valuable feedback. We're pleased to hear that you are happy with the revised manuscript.

- Regarding the generality of the architecture: thank you for adding the experiments with SocialGNN-V, I believe this makes a much more compelling case for the generality of the model.

Thank you for your support.

- Regarding more finegrained analysis: thank you for adding the RSA analysis! I think this is really informative and makes the claims much stronger. As a way to make the

particular claim about SocialGNN and SIMPLE being complimentary, it could be really interesting to try a hybrid model that uses a weighted combination of their predictions to see if that can explain human performance even better than either alone. (This would be, I think, somewhat more compelling than the semi-partial correlation analysis, though that already I think is helpful).

We agree, exploring the performance of hybrid models would be a more direct way to understand the relation between these two models and is an important avenue to pursue in future work.

- Regarding the term "visual": thank you for updating the paper to be more precise in the terminology, I think it is much clearer now. Though just to remark on one comment: "we believe... this is largely a solved computer vision problem". As someone working in machine learning, it is definitely not! I agree simple solutions might work well in the PHASE dataset so perhaps it could be considered "solved" in that case, but I'd be very careful about claiming too much about what computer vision models can do in the general case without actually demonstrating it.

Thank you for raising this concern. We agree that particularly in natural videos, these are still major challenges. We apologize for the imprecise language in our last reviewer response. We did not use this language anywhere in the paper.

- Regarding the RNN: thank you for clarifying that it is a pre-existing model, and for performing the additional fine-tuning to fix its performance. From the ML perspective though, I still think it is a strange architecture and not one that would work well, so it still feels to me like a strawman baseline and that more meaningful baselines could be devised. But, I am glad to see the addition of the CNN baseline and am satisfied with that addition.

Thank you for this perspective. We agree it is helpful to have a more standard CNN baseline. We are glad that you are satisfied with these additions.

- Regarding the SocialGNN architecture: that's fair to have chosen the architecture for controlled manipulation. I would encourage you though in future work to look at more canonical and battle-tested GNN architectures, such as GCN (graph convolutional networks) or MPNN (message-passing neural network) as I mentioned. In particular, using a recurrent MPNN would address the difficulty you alluded to earlier in the rebuttal: "We did not implement SocialGNN-E in the gaze dataset because the human annotations included in the dataset were collected for each person in each video. Since

the number of people can vary video to video it is not straightforward to convert these to a fixed number of edge ratings.". MPNNs have no requirement for a fixed number of edges.

Thank you for these references. We agree they are important benchmarks to consider in future work.

- Thank you for the remaining clarifications and edits.